



# The mechanism of sapropel formation in the Mediterranean Sea: Insight from long duration box-model experiments

Jan Pieter Dirksen[1] and Paul Th. Meijer[1]

[1]Utrecht University

**Correspondence:** Jan Pieter Dirksen (j.p.dirksen@uu.nl)

**Abstract.** Periodic bottom water oxygen deficiency in the Mediterranean Sea has led to the deposition of organic rich sediments during geological history, so called sapropels. Although a mechanism linking the formation of these deposits to orbital variability has been derived from the geological record, physics-based proof is limited to snapshot and short time-slice experiments with (Oceanic) General Circulation Models. Specifically, previous modelling studies have investigated atmospheric and
oceanographic equilibrium states during orbital extremes (minimum and maximum precession).

In contrast, we use a conceptual box model that allows us to focus on the transient response of the Mediterranean Sea to orbital forcing and investigate the physical processes causing sapropel formation. The model is constrained by present day measurement data, while proxy data offers constraints on the timing of sapropels.

The results demonstrate that it is possible to describe the first order aspects of sapropel formation in a conceptual box model.
A systematic model analysis approach provides new insights on features observed in the geological record, such as timing of sapropels, intra-sapropel intensity variations and interruptions. Moreover, given a scenario constrained by geological data, the model allows us to study the transient response of variables and processes that cannot be observed in the geological record. The results suggest that atmospheric temperature variability plays a key role in sapropel formation, and that the timing of the midpoint of a sapropel can shift significantly with a minor change in forcing due to nonlinearities in the system.

## 1   Introduction

### 1.1   Background

The response of ocean circulation to changes in atmospheric forcing is an important element of the climate system. Using computer models applied to the geological past we can exploit the sedimentary record of variation in circulation for mechanistic insight. The Mediterranean Sea is of particular interest, as abundant and exceptionally well dated proxy data and present-day
measurement data is available and it is a basin that displays processes such as thermohaline circulation and gateway control that play a role on the global scale as well.

Presently, the Mediterranean Sea is an evaporative basin (Romanou et al., 2010) with a small annual mean heat loss to the atmosphere (Song and Yu, 2017). Water from the Atlantic flows in to the Mediterranean Sea at the Strait of Gibraltar and is then



subjected to buoyancy loss due to evaporation and cooling. This results in the formation of intermediate water in the Levantine
basin which spreads throughout the basin (Hayes et al., 2019; Wu and Haines, 1996).

During winter, in parts of the basin situated at relatively high latitude, cold and dry winds induce a further density increase which may lead to the formation of deep water (Schroeder et al., 2012). Specifically, deep water formation (DWF) occurs over the shallow northern Adriatic Sea (Malanotte-Rizzoli, 1991) and the Aegean Sea (Gertman et al., 2006; Roether et al., 1996) and in the form of open-ocean deep convection in the Gulf of Lion (Marshall and Schott, 1999) and the southern Adriatic Sea
(Bensi et al., 2013). Dense water formed in the Adriatic and Aegean Seas, both marginal basins of the Mediterranean Sea, flows out over the seafloor into the deeper parts of the main basin.

The basin's semi-enclosed nature causes the system to be very sensitive to climatic perturbations and the geological record holds an expression of this sensitivity in the form of the regular occurrence of organic rich deposits, known as sapropels (Rossignol-Strick, 1985; Rohling et al., 2015; Hilgen, 1991; Lourens et al., 1996; Cramp and O'Sullivan, 1999). Sapropels
are thought to form when freshwater input increases as a response to enhanced East African summer monsoon activity during precession minima (Rossignol-Strick, 1985; Rohling et al., 2015). The low density fresh water then forms a lid at the surface, stopping or reducing the strength of the overturning circulation. This hypothesis can be nuanced by noting that the Nile does not enter the basin at a DWF site, but rather close to the location where intermediate water forms. A large part of the DWF involves this intermediate water (Schroeder et al., 2012). Reducing the density of the intermediate water implies a decrease or
absence of a (positive) vertical density gradient, also diminishing or stopping the formation of deep water. In contrast, run-off into the marginal basins directly affects the buoyancy at the DWF sites. For deep convection in for example the Levantine basin (which can happen with present day conditions, Gertmann et al., 1994) a decrease in surface water density directly decreases or stops DWF. With decreasing DWF, the supply of oxygen to the deep water diminishes, potentially causing anoxia and the preservation of organic matter in the Eastern Mediterranean Sea. Moreover, nutrient input increases with river outflow as
well, thereby affecting primary production, export of organic carbon to the deep water and, consequently, oxygen consumption (Calvert et al., 1992; De Lange and Ten Haven, 1983; Thomson et al., 1999; van Helmond et al., 2015; Weldeab et al., 2003).

In this paper we present a simple three box model of the Mediterranean Sea, which includes most mechanisms commonly invoked to explain sapropel formation described above. With the model we study which processes determine when and why sapropels form the way they do. Our aim is to gain a new perspective on the timing of the sapropel, relative to the forcing, as a
significant part of the late Neogene geological time scale depends on this relation (Hilgen et al., 1995; Krijgsman et al., 1999) and views on the timing of the mid point are contested in more recent publications(Channell et al., 2010; Westerhold et al., 2012, 2015). A low complexity model allow us to do long runs and explore the parameter space to a much greater extent than high complexity models. Long runs are necessary to study the transient response of the system over a full precession cycle.

As described in modelling studies (such as Marzocchi et al., 2015) as well as in observational studies (for example Herbert
et al., 2015), surface air temperatures have also been found to vary over a precession cycle, where precession minima are estimated to have been 1-3 $^{\circ}C$ warmer (annual average) than precession maxima. Since heat loss depends on the temperature difference between the water surface and atmosphere, this is another factor that decreases buoyancy loss during precession





minima. We will examine the relative importance of this effect by running the model both with, and without atmospheric temperature variability.

## 1.2 Previous modelling studies

Just like the Last Glacial Maximum, the time of sapropel formation has been recognized early on in the application of OGCMs to Mediterranean circulation, as a configuration that makes for an interesting contrast to the present-day state (Bigg, 1994; Myers et al., 1998; Myers and Rohling, 2000; Myers, 2002; Meijer and Tuenter, 2007; Meijer and Dijkstra, 2009); and more recently, using a more advanced model, (Mikolajewicz, 2011; Adloff et al., 2011). Several studies have explored the coupling of circulation models to models of the biogeochemical cycling, first offline and then in truly combined fashion (Stratford et al., 2000; Bianchi et al., 2006; Grimm et al., 2015). All these studies have in common that they are limited to time spans much shorter than the precessional cycle. The only previous box models related to the sapropel problem are those by Matthiesen and Haines (2003) and Amies et al. (2019), but these models lacks a representation of the deep waters of the basin.

## 2 Methods

### 2.1 Model set-up

The Mediterranean Sea is represented by three boxes in our model: the high latitude marginal basins (intermediate and surface water, box 1), the open Mediterranean (surface and intermediate water, box 2) and the deep water (box 3) (see Fig. 1). Box 1 and 2 have fluvial input (sourced from boxes R1 and R2, see Fig. 1) and exchange with the atmosphere (represented by boxes A1 and A2, see Fig. 1). The surface forcing is further explained in subsection 2.2. Each box has its own temperature and salinity. Boxes 1 through 3 are dynamic: the temperature, salinity and density is calculated during each time step, based on the incoming and outgoing salt and heat. The Atlantic, both rivers and both parts of the atmosphere can be seen as static boxes: their salinity, temperature and density are constant.

Circulation is modelled by including downward, vertical fluxes, of which the magnitude depends on the density difference between the surface/intermediate layer and the deep water (D1 and D2 in Fig. 1). This DWF is driven by buoyancy loss, due to evaporation (e1 and e2 in Fig. 1) and heat exchange with the atmosphere, which is modelled as a relaxation (i.e. the surface water temperature relaxes to the temperature of the associated atmosphere box, fluxes I1 and I2 in Fig. 1). Note that the DWF in box 1 captures the behaviour of the marginal basins of the Eastern Mediterranean sea, but is also an approximation of the open ocean convection in the Gulf of Lyon (see subsection 1.1). In the typical situation that the Mediterranean surface/intermediate water at the Strait of Gibraltar is more dense then the Atlantic water and E-P-R is positive (net evaporation), it is the outflow to the Atlantic (Qo in Fig. 1) that depends on the density difference between the adjacent water masses. The inflow into the Mediterranean Sea is then the sum of the outflow to the Atlantic and the net evaporation (E-P-R). The equations used in the model are further explained in subsection 2.3.





In addition to the water fluxes, diffusive mixing is also included in the model. In contrast to the water fluxes, no net water transport occurs as a result of the mixing. Rather, properties are exchanged between adjacent boxes. The amount of horizontal mixing (between the upper boxes) is constant, while the vertical mixing is a function of the density difference between the boxes in question.

A first order approximation of deep water oxygen concentration is included in the model to get a better understanding of when oxygen deficiency occurs. The oxygen concentration of the upper boxes is assumed to be in equilibrium with the atmosphere and is therefore constant. The oxygen concentration of the deep water (box 3) depends on the deep water fluxes, mixing and oxygen consumption. Oxygen consumption depends on river outflow, as a first order approximation of the nutrient input, and oxygen concentration. When the oxygen concentration drops below a threshold level, aerobic respiration is assumed to stop, reducing the oxygen consumption to a background level. The use of constant volume for the boxes implies (i) that we take there to always be a distinction between surface/intermediate and deep cell, and (ii) that the upper cell always extends to the same depth. The upper cell appears to be set up by the exchange with the ocean (see Meijer and Dijkstra, 2009 for the Mediterranean Sea and Finnigan et al., 2001, for a generic buoyancy-driven marginal sea) and is likely a persistent feature of Mediterranean circulation as long as there is an exchange flow. Moreover, starting from a state that does have DWF and a separate deep cell, OGCM experiments of reduced net evaporation show a halting of deep circulation while keeping the upper cell more or less in place (Meijer and Dijkstra, 2009).

In the present-day Mediterranean Sea DWF is the last step in a chain of processes (See the introduction). Our model does not include the intra-annual variability, and the basin geometry is only represented in abstract form. However, in the sense that the model does capture both the effect of salinity increase and temperature decrease on upper-water density it is expected to form a fair representation, qualitatively speaking, of the essence of the overturning circulation. To which extent this is true will have to follow from more advanced models. Note that the model of Matthiesen and Haines (2003) also neglects the seasonal cycle. During winter, convection occurs (Schroeder et al., 2012) and the depth of the intermediate water is relatively stable. We can therefore reduce the conceptual model to an open surface/intermediate box, a marginal surface/intermediate box and a deep water box, all with constant volumes. While the formation of deep water it self is a seasonal process, we parametrize the seasonal variability by calculating an annually averaged DWF flux. To do so, perpetual winter conditions have to be assumed, as deep water would not form with annual average conditions, while we know that DWF occurs during present winters.

## 2.2 Surface forcing

The transient response of circulation and water properties to precession induced climate change is modelled by altering the evaporation and river outflow for each box at every time-step. The analyses presented in this paper all use sine-waves to force the model, but any temporal variation could be used, such as that of the insolation curve. To be precise, the model forcing used in this paper is derived from a normalized sine-wave with a 20 kyr period, to reflect climatic precession. The amplitude and offset is then altered for evaporation and fluvial discharge in boxes 1 and 2. The phase of evaporation relative to the precession forcing is uncertain (see subsection 3.4) and is therefore varied between runs. The phase of the river discharge is kept at 0 degrees.





The fluvial discharge in box 2 is interpreted as the Nile outflow and other run-off from Africa. Prior to the construction of the Aswan High Dam in 1964, average Nile discharge was $2.7 \cdot 10^3 \ m^3/s$ (Rohling et al., 2015). Present day run-off from Africa is approximately $1.4 \cdot 10^3 \ m^3/s$ (Struglia et al., 2004). A recent modelling study (Amies et al., 2019) suggests that peak run-off

from Africa may have been 8.8 times larger than present during sapropel S5, note that this model does not consider changes in outflow from Europe.

Fluvial discharge in to the high latitude marginal basins of the Mediterranean Sea (R1 in the model) is presently approximately $6.7 \cdot 10^3 \ m^3/s$ (Struglia et al., 2004) . Increased runoff from Europe into the eastern Mediterranean has been proposed as a possible source for extra fresh water during precession minima (Rossignol-Strick, 1985; Rohling et al., 2002; Scrivner

et al., 2004)

The current net evaporation (E-P) is approximately $0.9 \ m/yr$ (Romanou et al., 2010). During sapropel times, net evaporation is hypothesized to have decreased (Rohling, 1994), although this has not been quantified. We therefore test a broad range of net evaporation, from $0.2$ to $2 \ m/yr$ to accommodate for these uncertainties.

## 2.3 Model equations and parameters

Here we first discuss the flux equations resulting from the model set-up and assumptions described above, followed by the equations used to integrate all flux-equations into a fully functioning model. All parameters are given in Table 1.

Observational and modelling studies (Herrmann et al., 2008; Schroeder et al., 2012) have shown that during colder winters, more deep water is formed. Hence, it makes sense that the magnitude of the vertical, downward fluxes ($D_1$ and $D_2$, see equations 2 and 3) depends on oceanographic (and thereby indirectly also atmospheric) conditions. The most simple way of

implementing this behaviour on a yearly resolution, is to assume a linear relationship between the density difference and flux magnitude (similar to Matthiesen and Haines (2003)). When the density of the overlying water mass is smaller than that of the deep water, the water column is stratified and no vertical flux exists. To clip negative components of a flux to 0, we use the form $F_{j,i} = max(0, a)$, where $a$ is the flux in question. We therefore define the following mathematical operator:

$$max(a, b) = \begin{cases} a & \text{for} \quad b \leq a \\ b & \text{for} \quad b > a \end{cases} \tag{1}$$

The proportionality of DWF to surface to deep water density difference is determined by a efficiency constant, $c_{13}$ and $c_{23}$ for $D_1$ and $D_2$ respectively. The magnitude of these constants is chosen in such a way that a realistic deep water flux occurs at a present-day density difference. In the current circulation, $D_2$ does not occur annually, making it difficult to determine $c_{23}$ empirically. By assuming that the DWF process in box 2 is the same as in box 1, $c_{23}$ can be taken as 4 times larger than $c_{13}$, proportional to the difference in surface area of boxes 1 and 2. We therefore define the DWF fluxes in equations 2 and 3.

$$D_1 = max(0, c_{13} \cdot (\rho_1 - \rho_3)) \tag{2}$$





$$D_2 = max(0, c_{23} \cdot (\rho_2 - \rho_3)) \tag{3}$$

At the Strait of Gibraltar, the exchange has two components from which the in- and outflow is calculated (see equations below): a density driven flux $Q_o$ (Eq. 4) and a compensating flux $Q_i$ (Eq. 5). The magnitude of $Q_o$ has a square-root relation

to the horizontal density difference at the strait, in accordance with (Bryden and Kinder, 1991) . Theoretically, this flux should be able to change direction, when the density difference changes sign. We therefore multiply the square-root of the absolute value of the density difference with the sign of the density difference. Note that the direction of the fluxes (i.e. whether it goes in or out of the Mediterranean Sea) is determined in equations 13 and 12. The conductivity parameter $c_{20}$ is again calibrated on present-day conditions (Schroeder et al., 2012; Jordà et al., 2017; Hayes et al., 2019). The compensating flux $Q_i$ can then

be calculated as the difference of $Q_o$ and the total freshwater budget of the Mediterranean Sea, to allow for conservation of volume.

$$Q_o = \begin{cases} -c_{20} \cdot \sqrt{|\rho_2 - \rho_0|} & \text{for} \quad \rho_2 \leq \rho_0 \\ c_{20} \cdot \sqrt{|\rho_2 - \rho_0|} & \text{for} \quad \rho_2 > \rho_0 \end{cases} \tag{4}$$

$$Q_i = Q_o - R_1 - R_2 + e_1 + e_2 \tag{5}$$

Heat exchange with the atmosphere depends on the temperature gradient with the surface water, thereby relaxing the temperature of the surface box to that of the atmosphere, similar to Ashkenazy et al. (2012). The heat exchange therefore is calculated by multiplying the temperature difference between the atmosphere and water box by a relaxation parameter. This relaxation parameter depends on the density of the water. In the model we rewrite this to an equivalent volume flux, $H_1$ and $H_2$ (in $m^3/s$), so that they can be treated as volume fluxes in the calculation of $d\boldsymbol{T}/dt$. The constants that relate the temperature and

temperature gradient to a heat flux ($c_{1A1}$ and $c_{2A2}$ in equations 6 and 7) are chosen so that at present day temperatures, a heat flux of approximately $5\,W/m^2$ occurs, in accordance with Song and Yu (2017) and Schroeder et al. (2012). $c_p$ is the specific heat of water.

$$H_{1,A1} = \frac{c_{1A1}}{c_p \cdot \rho_1} \tag{6}$$

$$H_{2,A2} = \frac{c_{2A2}}{c_p \cdot \rho_1} \tag{7}$$

The equations above describe all fluxes driven by gradients. By combining these fluxes with the surface forcing, we can derive the other fluxes by assuming constant box volume:

$$F_{1,A1} = e_1 \tag{8}$$



$$F_{2,A2} = e_2 \tag{9}$$

$$F_{R1,1} = R_1 \tag{10}$$

$$F_{R2,2} = R_2 \tag{11}$$


$$F_{2,0} = max(0, -Q_i) + max(0, Q_o) \tag{12}$$

$$F_{0,2} = max(0, Q_i) + max(0, -Q_o) \tag{13}$$

$$F_{2,1} = max(0, F_{13} - F_{R11} + F_{1A1}) \tag{14}$$

$$F_{1,2} = max(0, -F_{13} + F_{R11} - F_{1A1}) \tag{15}$$

$$F_{1,3} = D_1 \tag{16}$$


$$F_{2,3} = D_2 \tag{17}$$

$$F_{3,2} = D_1 + D_2 \tag{18}$$

Mixing has a major impact on oceanic circulation, and must therefore be included in the model. Unlike the water fluxes
described above, mixing does not cause a net water transport between boxes, but rather an exchange of properties (salt, heat
and oxygen). In the model, we distinguish between horizontal and vertical mixing. Horizontal mixing, between boxes 1 and 2,
depends on a fixed length scale over which mixing occurs and diffusivity (see Eq. 19). Vertical mixing (see equations 20 and 21)





depends on the density difference between the boxes in question, where a larger density gradient causes more mixing. Thereby the diffusivity of vertical mixing effectively depends on the density difference. When the water column is stratified, mixing

does not stop completely, but rather decreases to a background level, representing the internal waves and other disturbances. In the model this is included by clipping the vertical mixing to a fixed level ($k_{bg}$) when the density gradient becomes very small or negative.

$$m_{1,2} = k_{12} \cdot L \tag{19}$$

$$m_{1,3} = max(k_{bg}, (\rho_1 - \rho_3) \cdot k_{str} + k_{bg}) \cdot Wc_1 \tag{20}$$

$$m_{2,3} = max(k_{bg}, (\rho_2 - \rho_3) \cdot k_{str} + k_{bg}) \cdot Wc_2 \tag{21}$$

Oxygen is supplied to the deep water from the surface by fluxes $D_1$ and $D_2$ (equations 2 and 3) as well as through mixing with boxes 1 and 2 ($m_{1,3}$ and $m_{2,3}$, equations 20 and 21). The oxygen concentration in boxes 1 and 2 is assumed to in equilibrium

with the atmosphere and therefore constant. For the deep water, three oxygen consumption regimes are defined: when the oxygen concentration is above $60 \, \mu M$, the deep and benthic fauna is assumed to be stable and biological plus abiotic oxygen consumption is set to $0.4 \, \mu M$ in total per year (corresponding to $1 * 10^{12}$ mol/yr, following Powley et al. (2016). River outflow also affects oxygen consumption in this regime. With a deep water oxygen concentration between 60 and $0 \, \mu M$, only abiotic oxygen consumption occurs, which is set to $0.2 \, \mu M$ per year. When the deep water is completely anoxic, oxygen consumption

stops as well. Other processes affecting sapropel formation, such as nutrient dynamics and increased productivity due to the development of a deep chlorophyll maximum (Rohling et al., 2015; De Lange et al., 2008; Kemp et al., 1999; Rohling and Gieskes, 1989; Slomp et al., 2002; Van Santvoort et al., 1996; Santvoort et al., 1997) are not explicitly included in the model, but are to some extent parametrized by the dependency of oxygen consumption on river outflow (see Eq. 22).

$$O_{consumption} = \begin{cases} max(0, (Oc_A + Oc_B + R_{tot} \cdot Oc_R)/dt & \text{for} \quad O > 60 \\ max(0, Oc_A) & \text{for} \quad O \leq 60 \end{cases} \tag{22}$$

The model parameters (excluding surface forcing) used in all runs are given in Table 1. Initial temperatures are set to $16 \, °C$ and salinities to 37 for all dynamic boxes; they have no effect on the outcome of the model runs after spin-up. With the strait efficiency used in this paper, the model has a typical equilibrium time of less then 1000 years, while a spin-up of 20 kyr is removed from the output. As the temperature in boxes 1 and 2 relaxes to both the Atlantic temperature and the air temperatures, TA1, TA2 and T0 effectively set the temperature range of boxes 1, 2 and 3. The river inflow also affects temperature, but has a

much smaller impact due to the relatively small amount of water (2 orders of magnitude smaller than the Atlantic exchange). The air temperatures are chosen as winter values, since average air temperatures do not result in a realistic atmospheric heat loss and DWF. The temperature of the river water does not have a large influence on the model outcome.





The numerical integration of the model is best thought of as a matrix-vector representation. This same set-up could be used for an arbitrary configuration (and number) of boxes, to represent different oceanographic settings. The volumes of the boxes

are calculated from the depth and surface area for all dynamic boxes:

$$V = A \cdot d \qquad (23)$$

Except for a flux from box 3 to box 1, water can flow between all boxes in both directions. In the model, three types of fluxes exist: predefined fluxes, density driven fluxes and balancing fluxes. The predefined fluxes are used to force the model: evaporation and river discharge. The density driven fluxes are the DWF (unidirectional) and, depending on the sign of the density

difference, Atlantic inflow or Mediterranean outflow. All other fluxes are of such magnitude that volume is preserved. During each time-step in the model, the salinity, temperature and density for the next time-step are calculated from the fluxes and mixing. In the model script, these equations are only defined for dynamic boxes, increasing the model efficiency significantly. Below the equations are given in matrix form. The volumes of static boxes are infinite, as their temperature and salinity do not change, regardless of in- and outgoing fluxes. Note that the atmosphere boxes are the last two boxes in matrix $\mathbf{F}$ (boxes $n$ and

$n-1$).

The matrix $\mathbf{G}$ describes the water fluxes for the calculation of the new temperature, where $\mathbf{I}$ is the identity matrix and $l$ the unit vector.

$$\mathbf{G} = \mathbf{F} + \mathbf{I} \cdot \sum_j \mathbf{F}_{ij} \cdot l_i \qquad (24)$$

The matrix $\mathbf{P}$ describes the water fluxes for the calculation of the new salinity, where $\mathbf{J}$ is a matrix of ones. The only difference

with $\mathbf{G}$ being that evaporation is excluded (since evaporated water does not contain salt).

$$\mathbf{P} = \mathbf{F} + \mathbf{I} \cdot (\mathbf{J}_{n,1} - (l_n + l_{n-1})) \cdot \sum_j \mathbf{F}_{ij} \cdot l_i \qquad (25)$$

The matrix $\mathbf{N}$ describes the mixing fluxes for the calculation of both the new temperature and salinity.

$$\mathbf{N} = \mathbf{M} + \mathbf{I} \cdot \sum_j \mathbf{M}_{ij} \cdot l_i \qquad (26)$$

$W$ is a vector so that $W_i = \frac{1}{V_i}$.

Similar to $\mathbf{F}$, the heat fluxes (equations 6 and 7) are placed in $\mathbf{H}$, which is of the same size as $\mathbf{F}$ and where all undefined elements are zero. Then the change in temperature for each time-step equals:

$$T(t+1) = T(t) + (\mathbf{G} + \mathbf{N} + \mathbf{H}) \cdot T(t) \cdot W \qquad (27)$$

and for salinity:

$$S(t+1) = S(t) + (\mathbf{P} + \mathbf{N}) \cdot S(t) \cdot W \qquad (28)$$





The density for the next time step is calculated from the temperature and salinity using the EOS80 formula (on Oceano-graphic Tables, 1986). Note that vectors $V, S, T$ and $\rho$ include both static and dynamic boxes.

The deep water oxygen concentration of the next time step is similarly:

$$O(t+1) = max(0, O(t) + (\mathbf{F} + \mathbf{M} - \mathbf{N} - O_{consumption}) \cdot O(t) \cdot W) \tag{29}$$

Note that the oxygen concentration is only calculated for the deep water (making $O$ and $O_{consumption}$ effectively scalars) and that the surface water boxes have a constant oxygen concentration.

## 3 Analysis and results

### 3.1 Reference experiment

In the reference experiment the sine functions for the forcing are calibrated such that the precession maximum corresponds to present-day values—given that the orbital configuration is close to a precession maximum today. The curves are shown in Fig. 3A. All runs use a spin up of a full precession cycle, which is excluded from the figures and analyses; model run time T (horizontal axes) is set to 0 at the end of the spin up. All figures show an entire precession cycle, with the precession maxima falling at T=0 and T=20 kyr. The precession minimum sits at T=10 kyr.

Nile outflow increases from $5 \cdot 10^3$ to $3 \cdot 10^4$ $m^3/s$, while river outflow from Europe only increases from $5 \cdot 10^3$ to $1.2 \cdot 10^4$ $m^3/s$ and evaporation decreases from $0.9$ to $0.75$ $m/yr$. While quantitative reconstructions of fluvial discharge and evaporation during sapropel formation are not available, these minimum and maximum values are in agreement with Marzocchi et al. (2015). All other parameters, found in Table 1, do not vary with time.

From Time=0 towards the precession minimum, the river outflow increases and, as a result, salinities decrease, as shown in Fig. 3B. After the precession minimum river outflow decreases again and salinities increase. The differences in salinity between the boxes decreases towards the precession minimum, and increases again after the precession minimum. The amplitude of the salinity variability is much smaller in the open Mediterranean box (Box 2), as it is connected to the Atlantic (Box 0), which has a constant salinity in this run. Deep water salinity (Box 3) lags the salinity of the upper boxes (this will be interpreted after describing the other graphs). As a result of this behaviour, the salinity of the marginal box (Box 1) briefly drops below the deep water salinity just prior to the precession minimum.

The temperatures, shown in Fig. 3C, do not change drastically, except for a decrease in temperature at the margins in the interval surrounding the precession minimum (we will come back to this below).

As temperature does not change much, density variability (Fig. 3D) is largely determined by changes in salinity. The dip in marginal temperature has an opposite effect on density compared to the salinity fluctuation, consequently the decrease in surface to deep density gradient is relatively small, and the marginal density does not drop below the deep water density.

Nevertheless, the decrease in the vertical density difference causes a decrease in DWF (D1 in Fig. 3E, also see Eq. 2). DWF in the open Mediterranean box (D2) does not occur in this run, since the density in the upper open Mediterranean box never exceeds the density of the deep water box. The cause of the previously mentioned dip in marginal temperature, lies in the




reduction of DWF which in turn decreases the inflow of water from the open Mediterranean to the margins (Eq. 14). The decrease in supply of relatively warm water to the margins causes the water temperature of the margins to approach the much lower atmospheric temperature.

The outflow to the Atlantic (Qo, in Fig. 3D) depends on the density difference between the open Mediterranean and the Atlantic. Since the properties of the Atlantic water are kept constant in all presented model runs, the outflow only depends on the density of the open Mediterranean box. As expected then, the decrease in density of the open Mediterranean water in the interval surrounding the precession minimum causes a slight decrease in outflow to the Atlantic.

The deep water oxygen concentration (Fig. 3D) depends on 1) oxygen consumption, and 2) DWF and vertical mixing. When

the oxygen concentration is above 60 $\mu M$, it largely follows the same trend as DWF. Below 60 $\mu M$ oxygen consumption is much lower, causing it to not drop any further (see Eq. 22). Note that DWF does not have to stop completely to cause a decrease in the oxygen concentration; when the oxygen consumption combined with the out-flowing oxygen exceeds the supply of new oxygen, the deep water oxygen concentration decreases.

As we have seen in the description of Fig. 3D, the salinity decrease occurs in both the margins (where the deep water

forms in this run) and the open Mediterranean (where the water that flows to the margins originates from). The deep water salinity depends on DWF and mixing with the overlying boxes. Consequently, the deep water salinity always lags the salinity of the upper boxes. The amount of lag between the deep and the surface boxes depends on the water and property exchange with the deep box and is therefore not constant throughout the run. At the precession minimum, the lag is in the order 200 years. As the increase in river outflow towards the precession minimum reduces DWF and mixing, the lag between deep and

surface/intermediate water salinity also increases (too subtle to see in the graphs). As a result, there is a brief period, starting 1800 years before the precession minimum and ending 440 years after the precession minimum for the margins and 580 years for the open Mediterranean, where deep water salinity is higher than surface/intermediate water salinity. Because the changes in density largely depend on salinity in this run, and the dip in marginal temperature also slightly leads the precession minimum, it follows that the midpoint of this time interval of minimal DWF falls prior to the precession minimum.

The DWF does not stop completely in this run (see Fig. 3E), because the relatively warm open Mediterranean surface/intermediate water keeps the deep water warmer (through mixing) than the marginal water, see Fig. 3. This reference run highlights why the sapropel state is inherently transient: the DWF is only slowed down when the density of the upper boxes is decreasing, and increases again when the density starts to return to precession maximum conditions. Since density cannot decrease indefinitely, a state with minimum circulation cannot be maintained.

The deep water flux at the precession maximum is somewhat lower than found in observational data ($3 \cdot 10^5 m^3/s$ versus approximately ($1.6 \cdot 10^6\ m^3/s$), although comparable to the DWF one of the Eastern sub basins (Pinardi et al., 2015). Deep water oxygen is within error of the actual value (181 $\mu M$ in the model versus between 151 to 205$\mu M$ observed in the Western Mediterranean Sea and 160 to 219 $\mu M$ Eastern Mediterranean Sea, Powley et al., 2016). Other conditions, such as temperature and salinity match closely to present day winter conditions (as reported in Hayes et al., 2019). DWF only occurs at the margins

(box 1) in this run, the other deep water flux is only plotted for easy comparison to other runs (in which it does occur). None of the fluxes change direction in this run, resulting in relatively simple, although not entirely linear behaviour: the phase relation





between the salinities of the boxes is not constant, and the temperature of the marginal box, as well as the deep water oxygen curve are clearly not sinusoidal. We consider the period with minimal deep water oxygen concentration, $60\mu M$, to be the model equivalent of sapropel conditions. Although we only find a very short sapropel, this run demonstrates that the model (i)

is capable of approximating the present-day water properties and circulation when forced by present atmospheric conditions, and (ii) captures the reduction in DWF expected upon a change to wetter conditions.

## 3.2  Addition of atmospheric temperature variability

As described in the introduction, temperature variation due to precession likely also affected buoyancy loss. In order to examine this aspect, we run the model with a $3\,°C$ temperature increase at the precession minimum relative to the precession maximum.

For atmospheric box A1 the temperature increases from 12 to 15 $°C$, and for box A2 the temperature increases from 10 to 13 $°C$. Both air temperature curves are described by sine waves, as shown in Fig. 4C. We decide to maintain a constant temperature difference between the two atmospheric boxes as there is insufficient evidence for other options. All other parameters are set as described in the reference run.

The overall behaviour of the model is similar to that in the reference run, except that the temperatures of all boxes are now

higher during the interval surrounding the precession minimum (Fig. 4C). We still observe a minor decrease in marginal water temperatures at the precession minimum (cf. Fig. 3C), albeit much smaller than in the reference run, since it is now imposed on top of the trend caused by the changing atmospheric temperature. The net effect of a homogeneous basin wide temperature increase during the precession minimum is a further decrease in DWF during this time interval. We find a sapropel from t=2900 years to 6500 years, which therefore lasts 3600 years and the midpoint leads the precession minimum by 300 (see Fig. 4E).

When testing the parameter space, we find that changes in marginal and open Mediterranean air temperature have an opposite effect on DWF: when the air over the margins becomes warmer, heat exchange with the marginal water directly increases the buoyancy of the water involved in DWF, slowing the circulation down. An increase in open Mediterranean air temperature, in contrast, primarily affects the open Mediterranean surface/intermediate water, which mixes with the deep water over a large area. The resulting rise in temperature of the deep water lowers its density, and thereby increases the marginal to deep water

density gradient. Since this gradient controls DWF formation at the margin, an increase in open Mediterranean air temperature ultimately causes an increase in DWF. Since part of the open Mediterranean surface/intermediate water flows to- and mixes with the marginal water, the effect of the open Mediterranean air temperature increase on the margin-deep water density gradient is relatively small.

This run shows that an atmospheric temperature increases during the precession minimum significantly affects the duration

of sapropel conditions in the model. Since both observational and modelling studies find this temperature variability (Marzocchi et al., 2015; Herbert et al., 2015), it will be included in all following model runs.

## 3.3  Nonlinear behaviour

Next, we explore the effect of a transition to and from a time interval with a positive freshwater budget. Whether or not the freshwater budget of the Mediterranean Sea becomes positive during sapropel formation has been widely debated (Rohling,





1994, and references therein). Although our model cannot directly prove whether or not this has happened, it does allow us to study what the implications for the water properties and circulation would be, which should help in recognising the expression of a budget switch in the geological record. First we consider a scenario where only the freshwater budget of the margins becomes positive; in a subsequent run we force the model in such a way that the freshwater budget of the entire basin changes sign.

To have the freshwater budget of the margins become positive, the maximum outflow of river 1 is increased from $6.7 \cdot 10^3$ $m^3/s$ to $1.4 \cdot 10^4$ $m^3/s$ (Fig. 5A). All other parameters are kept the same as in the temperature variability run (Fig. 4).

  At a similar timing as the dip in temperature observed in the reference run, we now see a very large decrease in salinity at the margins from 9 to 13 kyr (see Fig. 5B). During this interval we observe that temperatures at the margins approach the temperature of the overlying atmospheric box, while deep water temperatures approach those found in the open Mediterranean

(see Fig. 5C). All are an experession of the disappearance of DWF at the margin (see Fig. 5D; elaborated below) which effectively stops the exchange of the margins with the rest of the basin. Conditions at the margins are mainly determined by the river input (causing low salinity) and atmospheric temperature. The properties of the deep water are now only determined by mixing with the open Mediterranean surface/intermediate box, and DWF in the same box, explaining the similar temperatures.

  When the salinity at the margins reach normal values again at 13 kyr, we observe a sudden subtle increase in deep water

salinity, due to the abrupt increase in DWF at the margin at this moment (see Fig. 5D).

  Because the change in salinity is much larger than the change in temperature, the densities of each of the boxes (Fig. 5D) behave similarly to the observed salinities seen in Fig. 5B.

  DWF at the margins is found to gradually decrease towards the precession minimum, then completely stop at around 8 kyr, and abruptly increase to normal circulation again at around 13 kyr. DWF in the open Mediterranean starts close to the

precession minimum and ends abruptly when DWF at the margins starts again. Deep water oxygen largely behaves as the total DWF, although it reaches a minimum before DWF stops completely and begins to increase only shortly after DWF in the open Mediterranean starts. Similar to previous runs, outflow to the Atlantic (Fig. 5E) is slightly lower during the precession minimum, because 1) the density difference between the Atlantic and open Mediterranean surface/intermediate box is smaller, and 2) the freshwater budget is closer to zero.

We thus find that when the freshwater budget in the marginal box temporarily becomes positive, DWF occurs in the open Mediterranean at the end of the low deep water oxygen interval (conditions associated with sapropel deposition), thereby terminating this interval early (as shown in Fig. 5). Deep water mixing with the much less dense water at the margins decreases the density of the deep water, thereby causing DWF in the open Mediterranean box. The result of this is a phase lead of the sapropel midpoint (as a result of the earlier termination), instead of a phase lag commonly reported in literature (Grant et al.,

390   2016).

  In the next run we force the model in such a way that the freshwater budget of the entire basin becomes positive during the interval straddling the precession minimum.



The maximum outflow of river 2 is set to $8 \cdot 10^4 \ m^3/s$ and the minimum evaporation to 0.74 m/yr (Fig. 6A), all other parameters are kept the same as in the temperature variability run. In the interval from approximately 9 to 13 kyr, the freshwater

budget of the entire basin reverses.

Salinities (Fig. 6B) decreases in response to the decrease in net evaporation. When the freshwater budget reverses, the exchange with the Atlantic decreases, causing less relatively saline water to flow into the upper boxes. Consequently, the salinity of the upper boxes further decreases. The deep water salinity only begins to decrease more when DWF at the margin starts again. When the freshwater budget becomes negative again, the salinities abruptly increase and then follow the freshwater

budget more or less linearly.

The main features of the temperature curves (Fig. 6C) are caused by the same events that are described above for the salinity variability, although temperature is also affected by heat loss to the atmosphere. Consequently, the same main features can be identified, with the difference that 1) the temperature of the upper boxes follows the air temperature curves, and 2) the amplitude is smaller, because the heat exchange with the atmosphere acts as negative feedback.

The changes in densities are predominantly determined by salinity, as the changes in temperature are relatively small in this run.

Reversing the freshwater budget also causes the density difference between the Atlantic and open Mediterranean surface/intermediate box to change sign. Consequently, the density driven flow goes from the Atlantic to the Mediterranean, instead of the other way around. In Fig. 6E, this is represented by the flux becoming negative. Note that this shift occurs almost

instantaneously.

In this run, we find a very sharp termination of the sapropel, followed by a brief period with lower oxygen concentration (as shown in Fig. 6). This is caused by a peak in DWF in both the margin and open Mediterranean when the freshwater budget changes sign. Just prior to the reversal of the freshwater budget, the density of the open Mediterranean surface/intermediate water is much lower than that of the Atlantic water. The reversal of the freshwater budget then causes a rapid increase in

surface/intermediate water throughout the basin, resulting in the peak in DWF.

The irregularities observed in all runs where the freshwater budget of (part of) the basin reverses all strongly depend on the model set-up.

### 3.4   Phase of evaporation

Recent modelling studies (Marzocchi, 2016) have shown that while evaporation and river outflow are both forced by preces-

sion, they may have a different phase relation to their forcing. To assess the effect of the phase of evaporation on sapropel formation, we calculate the sapropel midpoint and duration for a set of runs, with varying evaporation phase (all other parameters remaining unaltered between runs). Apart from the phase of the evaporation forcing, the model is forced exactly the same as in the atmospheric temperature variability experiment (as described in subsection 3.2).

As shown in Fig. 7, we find a maximum in sapropel duration when evaporation is almost in phase with precession. This is to

be expected, as minimum evaporation then coincides with maximum river outflow. Similarly, a minimum in sapropel duration is found when evaporation is almost exactly in anti-phase with precession. In between these peaks the sapropel duration as a





function of the phase of evaporation is described by a cosine. The sapropel midpoint is found to vary significantly (hundreds to thousands of years) when varying the phase of the evaporation forcing. The shift in sapropel midpoint relative to the precession minimum is at a maximum when evaporation lags or leads approximately 5 kyr. This makes sense, as the minimum in midpoint

shift occurs when evaporation is either in phase with precession, or in anti-phase (i.e. a shift of 0 or 10 kyr), the 5 kyr lead/lag falls right in between these points. The timing of the midpoint as a function of the phase of evaporation in between these extremes is described by a nearly perfect sine wave. Note that the peaks are not exactly at -5 and and 5 kyr, but slightly shifted, this is likely a result of the equilibrium time of the system.

This experiment, combined with the systematic testing of the parameter space, highlights that although the exact timing and

duration depend on the exact forcing, the minimum in deep water oxygen concentration always occurs close to the precession minimum and the model response is always quasi-linear, as long as fresh water budgets are not reversed.

We also find that the magnitude of the effect of the phase of evaporation on sapropel timing and duration depends on the amplitude of the evaporation variability (not shown). This makes sense, as the changes in circulation largely respond to freshwater budgets (the only difference between river inflow and evaporation being their respective temperatures) and the

amplitude of evaporation variability scales linearly with its impact on the variability of the freshwater budgets.

When systematically varying the components of the water budget within the limits mentioned in model setup, we find that in the regimes where the freshwater budget of (part of) the basin changes sign, sapropels are cut short considerably. When performing the same analyses described above, but now using the forcing of the first run in subsection 3.3, we find that this causes the midpoint of the sapropel to occur prior to the precession minimum (Fig. 8). Note that runs with multiple sapropel

intervals cannot be described as having a single midpoint or duration.

## 4 Discussion

### 4.1 Statistical analysis

One of the results of the model is that slight variations of forcing parameters can cause significantly different sapropel duration and timing. We therefore introduce a statistical test to determine the magnitude thereof, given the uncertainty of each of the

forcing variables. With eleven forcing parameters (the phase of evaporation and minima and maxima of R1, R2, TA1, TA2 and evaporation) it is not feasible to calculate all permutations at a meaningful resolution. We therefore randomly pick and run 200 permutations (fewer permutations would produce unreliable results), given the uncertainty of each parameter , and calculate the $1\sigma$ and minimum and maximum values of the resulting oxygen concentrations per time step (see Fig. A1 for an example). During testing, we can thereby visualize much more of the parameter space than when doing individual runs.

### 4.2 Model convergence

The annual resolution results from the concept that deep water forms during winter storms, making it the highest resolution possible as long as seasonal variability is not included. From a purely mathematical perspective however, the time resolution





should not affect the outcome significantly, as long as sufficiently small time step is used to prevent aliasing. We tested this by varying the temporal resolution given a certain forcing. We find that with a time step below 6 years, the only difference

in the model results (compared to the result when using a higher temporal resolution) is in the fluctuation around an oxygen concentration of $60~\mu M$, during sapropelic conditions. This fluctuation scales linearly with the time step, with an amplitude of approximately $0.4~\mu M$ at a time step of 1 year and $0.04~\mu M$ at a time step of 1/10th of a year, see Fig. 2. We conclude that a time step of 1 year is sufficient for the analyses in the study.

### 4.3 The role of assumptions and simplifications

All models require assumptions and simplifications to be made, as they are by design a simplified version of (part of) a system. Simple box models, such as the model presented in this paper, aim to identify the smallest subset of processes that can describe a certain phenomenon. As such, this model represents a generic semi-enclosed basin, given that no specific geometry is included. This also implies that by altering the parameter values and in some cases the strait exchange equations, the model can easily be adapted to other semi-enclosed basins, such as the Black Sea.

The model forcing used in this study is chosen to reflect either the variability described by the main hypothesis, or oceano-graphic and climatic variability deduced from modelling studies and the geological record as accurate as possible. Other processes such as the North Atlantic oscillation and solar activity are not taken into account, because they are not thought to be of first order importance for sapropel formation, as described in Rossignol-Strick (1985) and Rohling et al. (2015) for example. While these processes likely influence sapropel formation, they are unlikely to be essential.

The model output comprises an average value for deep water oxygen, as the deep water is a single box. In reality, however, oxygen concentrations vary spatially. A prime example of thereof is the absence of sapropels in most of the Western Mediterranean Sea. This abstraction should be taken into consideration when interpreting the model results. This model focusses on the transient response of water fluxes in the Mediterranean Sea, the oxygen output is calculated to get a first order impression of deep water ventilation. A biogeochemical model, comparable to the one presented in (Slomp and Van Cappellen, 2007), would

have to be included to specifically study bottom water oxygenation. We expect that the main difference with a biogeochemical model will be that in our model river input directly affects oxygen consumption, while the surface/intermediate boxes would act as a reservoir for nutrients (with their own feedbacks) in the biogeochemical model.

Note finally that in reality DWF occurs following two different mechanisms, as well as in multiple sub basins that each have their own conditions.

The regime in Fig. 5 relies on the freshwater budget of the margins changing sign. In reality there are many different marginal water masses in the sub-basins, rather than one single "margin". This makes it likely that the freshwater budget becoming positive in any one of these sub-basins will have similar consequences for the circulation. Since the freshwater budgets of these basins are independent, it would be possible to drastically alter the circulation multiple times during a single precession cycle. Presently, the Adriatic sea has a positive freshwater budget (Raicich, 1996), and the Aegean sea is known to have had a positive

fresh water budget in the past (Zervakis et al., 2004).



The simplicity of the model makes it especially suitable for describing transient, nonlinear behaviour, allowing for the identification of crucial mechanisms. More complex models, while providing other benefits, are generally too difficult to interpret on this level, or do not allow for runs of sufficient length to study the transient response over a full precession cycle. The presented model runs give an overview of the behaviour of the model. When systematically testing the parameter space, 495   we find that this behaviour largely depends on general trends and reversal of fresh water budgets, rather than specific forcing or parameter choices. This makes the results of the study much more robust and meaningful.

### 4.4   Describing nonlinear relationships and transient response

The occurrence of sapropels is often considered from a binary perspective: a sediment is either a sapropel, or it is not. The dominant forcing mechanism (astronomic variability), however, can be easily described by a combination of a limited number 500   of sine-waves: the resonant frequencies of the planetary bodies in our solar system (for example Laskar, 1988). For a single sapropel, only climatic precession—a nearly a perfect sine—is considered to be of first order importance in controlling bottom water oxygenation (Rossignol-Strick, 1985), with the rest of the orbital configuration mostly modulating the effect of precession. If a model strives to capture the current hypothesis of sapropel formation, starting from astronomic variability, it must therefore transform a sine wave into something that is not a sine wave, requiring the model to be nonlinear. Our model allows 505   for such behaviour. Even when considering intra-sapropel variability, thereby surpassing a binary approach, the sapropel record is clearly not sinusoidal (see for example Grant et al., 2016; Dirksen et al., 2019).

One of the main research questions of this study is when sapropel formation occurs. In a linear system, one would simply calculate the phase of the output with respect to the input. However, as the output is no longer linearly related to the input, this is not possible. A simple threshold analyses will not suffice either, as the cut-off level can have major impact on both 510   timing and duration, while a clear definition is not readily available. Furthermore, even when the threshold is defined, this method would not be usable for sapropelic marls, which are thought to be the result of the same process, but do not share the same chemical composition. We partly avoid this problem by instead considering the midpoint of the sapropel (when assuming a certain oxygen threshold, see subsection 4.5). While this can't be related directly to the sedimentary record, it does give insight into factors influence sapropel timing. Even this definition of sapropel timing becomes problematic when one or more 515   interruptions occur, since in that case there is more than one mid point.

We find that, when using realistic model forcing, stable sapropel conditions do not occur. Even when using constant forcing, a permanent complete stop of DWF either does not occur, or only under very specific conditions. Note that in the Black Sea permanent stratification does appear to occur, however, the positive freshwater budget allows some of the water flowing into the Black Sea at the Bosphorus Strait to sink to the deep water (Bogdanova, 1963), keeping it relatively saline. However, our 520   results indicate that sapropel conditions can occur transiently without a positive freshwater budget, with realistic forcing. We therefore conclude that studying the oceanographic state during sapropel conditions by modelling steady-state conditions with a stratified water column results in a very limited understanding of sapropel formation.



## 4.5 Comparison with geological data and other models

Comparing model results to geological data is most effective when an accurate age model is available for the geological data.We
will therefore only consider the five youngest sapropels in this paper. The most recent sapropel (S1) is thought to have been
triggered by sea level rise, which in turn resulted in a connection between the Black Sea and the Mediterranean Sea (Rohling
et al., 2015). As both sea level variability and exchange through the Bosporus Strait are beyond the scope of this paper, sapropel
S1 is not suitable for comparison. We therefore focus on sapropels S3, S4, and S5 in the rest of the section

In the run with variable air temperature (Fig. 4), the modelled sapropel duration and timing is within error with what has
been found for sapropel S3 and S4 in core LC21 (Grant et al., 2016). Note that the same study finds that sapropels S1 and S5
lag precession by 2.1-3.3 kyr. This suggests that our model is capable of capturing the most relevant mechanisms for S3 and
S4, but that other features not included in the model affected the timing of S1 and S5. For S5 there is evidence suggesting that
the Black Sea reconnected to the Mediterranean Sea within uncertainty of the onset of sapropel S5 (Grant et al., 2016, 2012;
Wegwerth et al., 2014)

It should be noted that while the model often shows a midpoint lag (relative to the insolation minimum) of a few hundred
years, uncertainties related to radiometric dating methods are often larger. However, we find that midpoint lag becomes larger
with decreasing strait efficiency, implying that during times with low sea level or otherwise restricted exchange, the lag might
become very relevant. A prime example of such a case would be the Messinian Salinity crisis and the surrounding intervals
(Topper and Meijer, 2015). Moreover, as shown in Figures 5 and 6, the alternative regimes can shift the midpoint of the
sapropel considerably, as shown in Fig. 8. Unlike the relatively minor shifts in midpoint resulting from only changing the
phase of evaporation, the resulting difference in sapropel timing is sufficiently large to be detectable in the geological record.

De Lange et al. (2008) find that the freshening of the surface waters starts earlier and lasts longer than the suboxic bottom
water conditions during S1. Our model also shows this behaviour in all of the presented runs (most notably in figures 3, 4 and
6). This makes sense, as the DWF does not stop completely when the surface water starts to become less saline; it only reduces
slowly. Furthermore, the oxygen has to be depleted for suboxic condition to occur, this is limited by oxygen consumption, and
further slowed down by vertical mixing and DWF. How much longer the period of reduced sea surface salinity lasts compared
to the period of suboxic bottom water likely depends on the exact location and water depth. The model is therefore mostly
useful to gain insight into the mechanisms, rather than the exact timing.

The regime described in Fig. 6 can show a sudden termination of the sapropel, which is similar to that seen in records of
sapropel S5 (Dirksen et al., 2019). In the model, such a sudden termination can only be achieved by forming deep water in open
Mediterranean, by reverse the freshwater budget of the entire Mediterranean. The coupling between the margins and deep water
is insufficient to cause such a sudden termination. This suggests that during S5 the freshwater budget of part of the basin, or
the whole basin potentially reversed. Such a large change in freshwater budget is in line with Bale et al. (2019), who found that
surface salinities in three different cores in the eastern Mediterranean were sufficiently low to support free-living heterocystous
cyanobacteria during sapropel S5. Moreover, oceanographic conditions may differ significantly between different parts of the
basin: the oxygen concentration (and related variables) a hypothetical core taken at the margin would be expected to show a





pattern more similar to the DWF in the marginal box, while a core in the open Mediterranean may be more similar to the open Mediterranean surface/intermediate water box.

Sapropel interruptions commonly occur in the stratigraphic record (for example in S5 in core LC21, Rohling et al., 2006, 2015). With slightly different settings, the sharp peak in deep water oxygen in Fig. 6E can be made to occur earlier and less intense, resulting in an interrupted sapropel. The model therefore suggests that such interruptions can occur without further external forcing. This hypothesis could be tested in the stratigraphic record by looking for evidence of a reversed freshwater budget of (part of) the basin during such interrupted sapropels, and constructing high resolution intra-sapropel age models to assess the relative timing of the relevant features compared to insolation.

Each sapropel in the geological record is different, this already becomes apparent when considering the first six: S1 has a different timing and is likely related to sea level variability (Grant et al., 2016; Hennekam, 2015). S2 is not found at all. S4 has an interruption in core LC21 (Grant et al., 2012), and the high resolution records of for example trace metals show very different characteristics. In the same core, the timing of the midpoint of S3 and S4 compared to insolation is the same, while that of S1 and S5 are different. S5 is much longer than all of the previous sapropels, does not have any burn down at at least one location (Dirksen et al., 2019), has an interruption in other locations (Rohling et al., 2006), and again shows generally different characteristics in different cores (Dirksen et al., 2019; Grant et al., 2016; Rohling et al., 2006). S6 again looks very different. We conclude that both in the geological record and in the model, a typical sapropel does not exist. The timing and mechanisms involved may differ considerably between sapropels and locations, as highlighted by our model results. Moreover, we find that an increase in freshwater budget alone is not sufficient to describe all key aspects of sapropel formation. An increase in atmospheric temperatures during the precession minimum (as observed in data and modelling studies Marzocchi et al., 2015; Herbert et al., 2015) directly affects buoyancy loss during the interval in which sapropels form. This makes atmospheric temperature variability an integral feature of the system, without it unrealistic evaporation or river outflow is needed to result in sufficiently sapropels. Our model results support this hypothesis.

## 5 Conclusions

The analysis presented in this paper illustrates that relatively simple models can give new, fundamental insights into the physical processes driving sapropel formation. The timing of sapropels relative to insolation has been widely studied in the sedimentary record. On the basis of our novel long-duration experiments we find that the timing of sapropels is very sensitive to the exact climatologic and oceanographic conditions.

The nonlinear response to insolation forcing implies that the sapropel record does not have a linear phase relation with insolation. The strongly nonlinear regimes in our model highlight that the mid-point of a sapropel can be shifted significantly with a minor change in forcing, by cutting it short with a sudden termination, while during the rest of the precession cycle the response can be very similar to the nearly linear regime presented in the reference experiment.



Our model results suggest that an increase in freshwater input alone, as in the general hypothesis for sapropel formation, does not provide a sufficient reduction in buoyancy loss to form sapropels as they are found in the geological record. We propose

590     that precession controlled atmospheric temperature variability also plays a key-role in the process of sapropel formation.





## Appendix A: Appendix A

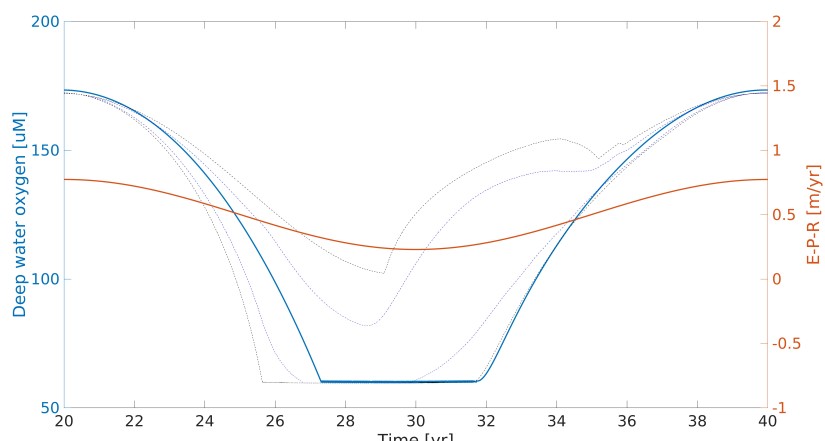

**Figure A1.** An example of the sensitivity analyses. Here the maximum of R1 is varied randomly by up to $5 \cdot 10^3 \ m^3/s$ above or below the general setting over 200 runs. The solid lines indicate the general run, the blue dashed lines indicate the upper and lower $1\sigma$ of each point in time, and the black dashed lines the minimum and maximum.

## A1

*Author contributions.* P. Meijer conceived the basic idea. J.P. Dirksen elaborated the concepts and performed the analyses. Both authors contributed to the model set-up and the writing of the manuscript.

595    *Competing interests.* The authors have no competing interests.

*Acknowledgements.* This work was financially supported by The Netherlands Research Centre for Integrated Solid Earth Science (ISES 2017-UU-23).





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





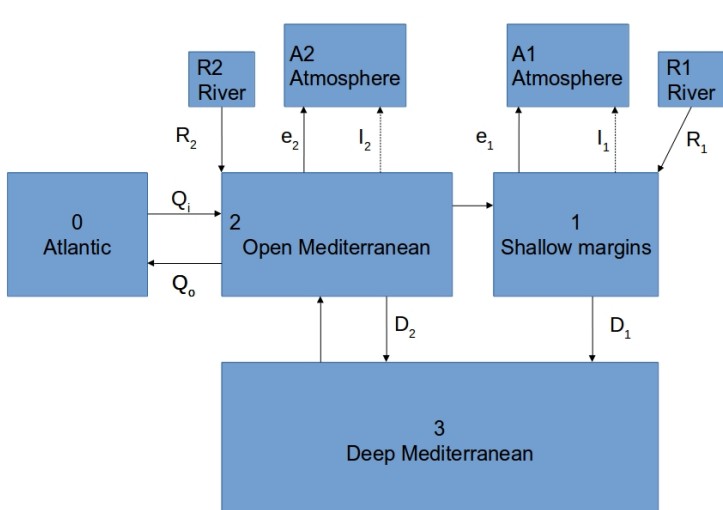

**Figure 1.** A schematic overview of the model set-up. The unlabelled fluxes are balancing fluxes, the equations for all fluxes are given in subsection 2.3. The direction of the arrows indicates the positive direction in the equations, all horizontal fluxes can change direction.





**Table 1.** All model parameters, excluding the model forcing.

| Name | Value | Units | Description |
|------|-------|-------|-------------|
| $c_A$ | 1.5 | $m/yr$ | Sea to air conductivity |
| $SFH$ | 0.1 | $W/m^2$ | Earth to sea heat flux |
| $c_{13}$ | $1 \cdot 10^6$ | $s^{-1}kg^{-1}$ | Conductivity between box 1 and 3 |
| $c_{23}$ | $4 \cdot 10^6$ | $s^{-1}kg^{-1}$ | Conductivity between box 2 and 3 |
| $c_{20}$ | $3.9 \cdot 10^5$ | $m^3/s/\sqrt{kg/m^3}$ | Conductivity at the Strait of Gibraltar |
| $k_{12}$ | $1 \cdot 10^{-4}$ | $m^2/s$ | Hor. mixing between box 1 and 2 |
| $L$ | 1000 | $m$ | Length scale of horizontal diffusivity |
| $k_{bg}$ | $4 \cdot 10^{-5}$ | $m^2/s$ | background vertical mixing strength |
| $k_{str}$ | $3.5 \cdot 10^{-4}$ | $m^5/(kg \cdot s)$ | vertical mixing strength/rho grad. |
| $A$ | $2.5 \cdot 10^{12}$ | $m^2$ | Surface area of the Mediterranean |
| $f$ | 0.2 | $-$ | Fraction of the surface area of box 1 |
| $d_1$ | 500 | $m$ | Depth of box 1 |
| $d_2$ | 500 | $m$ | Depth of box 2 |
| $d_3$ | 1000 | $m$ | Depth of box 3 |
| $S_0$ | 36.2 | $kg/m^3$ | Salinity of the Atlantic ocean |
| $T_0$ | 15 | $°C$ | Temperature of the Atlantic ocean |
| $T_{R1}$ | 16 | $°C$ | Temperature of R1 |
| $T_{R2}$ | 18 | $°C$ | Temperature of R2 |
| $O1$ | 230 | $uM$ | Surface water $O_2$ concentration |
| $Oc_B$ | 0.1 | $\mu M/yr$ | Biological oxygen consumption |
| $Oc_A$ | 0.1 | $\mu M/yr$ | Other oxygen consumption |
| $Oc_R$ | 0.1 | $\mu M \cdot s/m^3$ | oxygen consumption river outflow |
| $B_{th}$ | 60 | $\mu M$ | Threshold $O_2$ concentration |
| $shr$ | $4.187 \cdot 10^3$ | $J/(°K \cdot kg)$ | Specific heat of water |





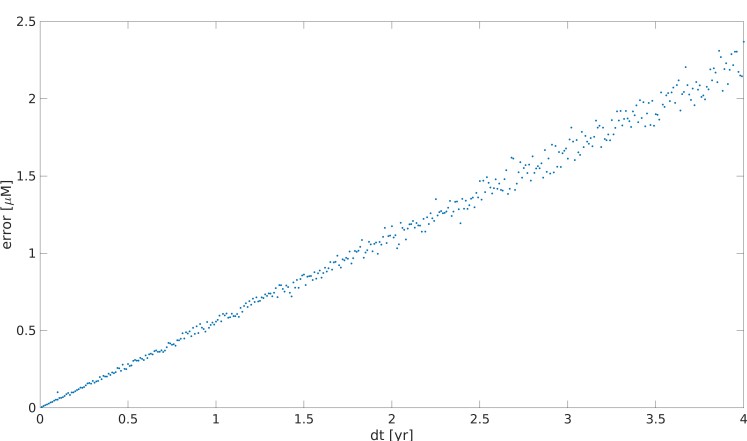

**Figure 2.** The variability of deep water oxygen over the interval where oxygen is below 60 $\mu M$ plotted against time step (each data point represents a model run).

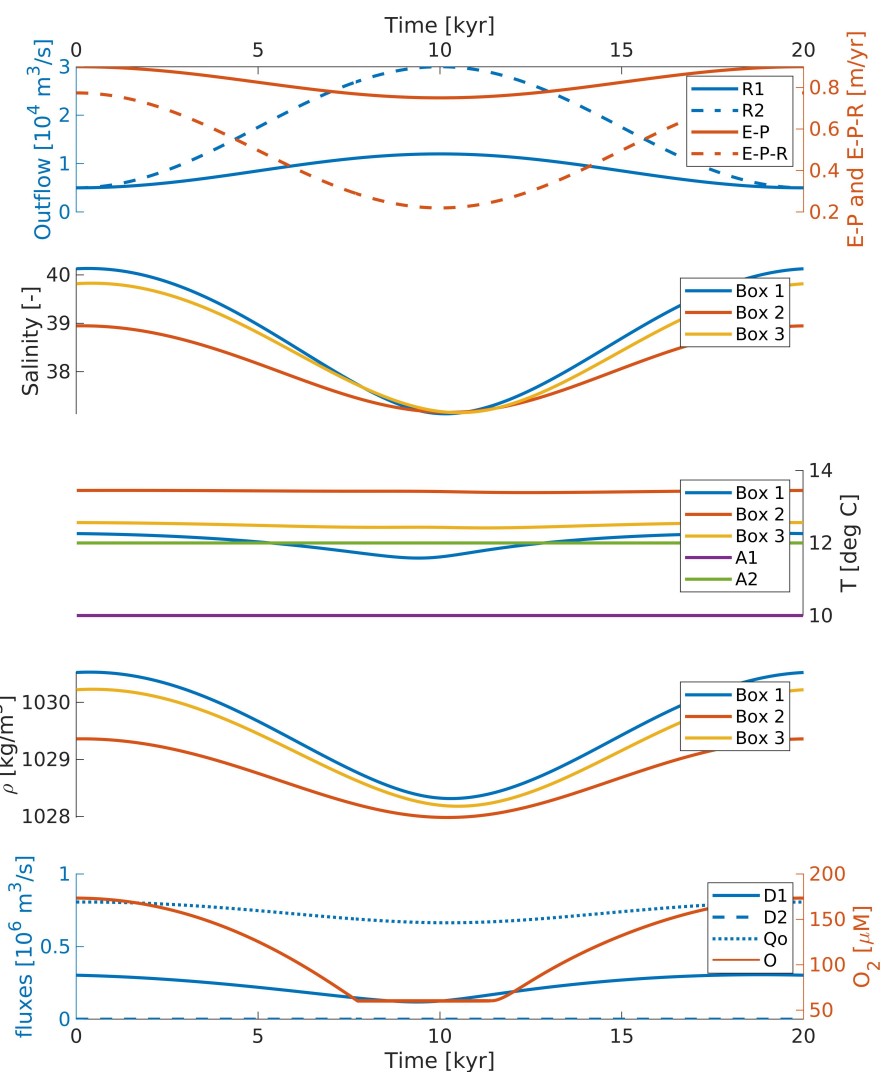

**Figure 3.** The forcing and results of the reference run. (A) The model forcing, with the river outflow on the left axis and the evaporation on the right axis. (B)-(D) For each box respectively the salinity, temperatures, and densities. (E) The relevant fluxes (left axis) and the deep water oxygen concentration (right axis)





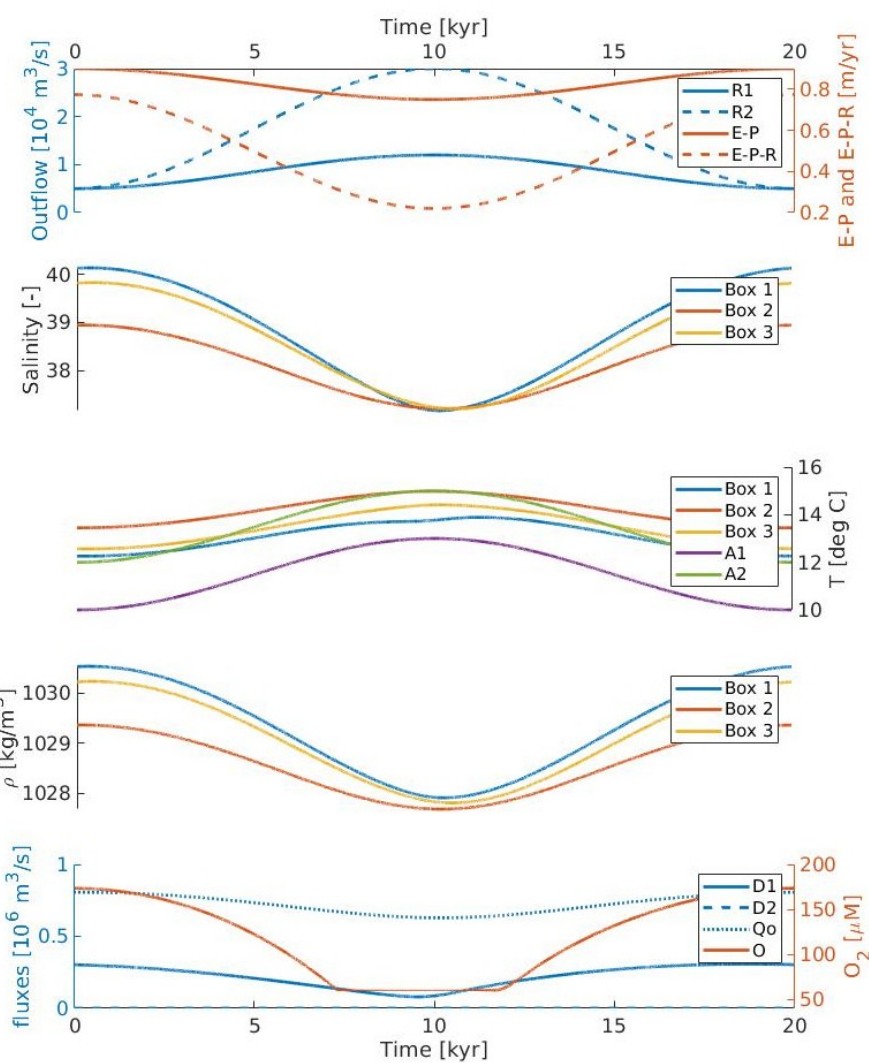

**Figure 4.** The forcing and results of the temperature-variability run. Layout of the panels is the same as in Fig. 3.

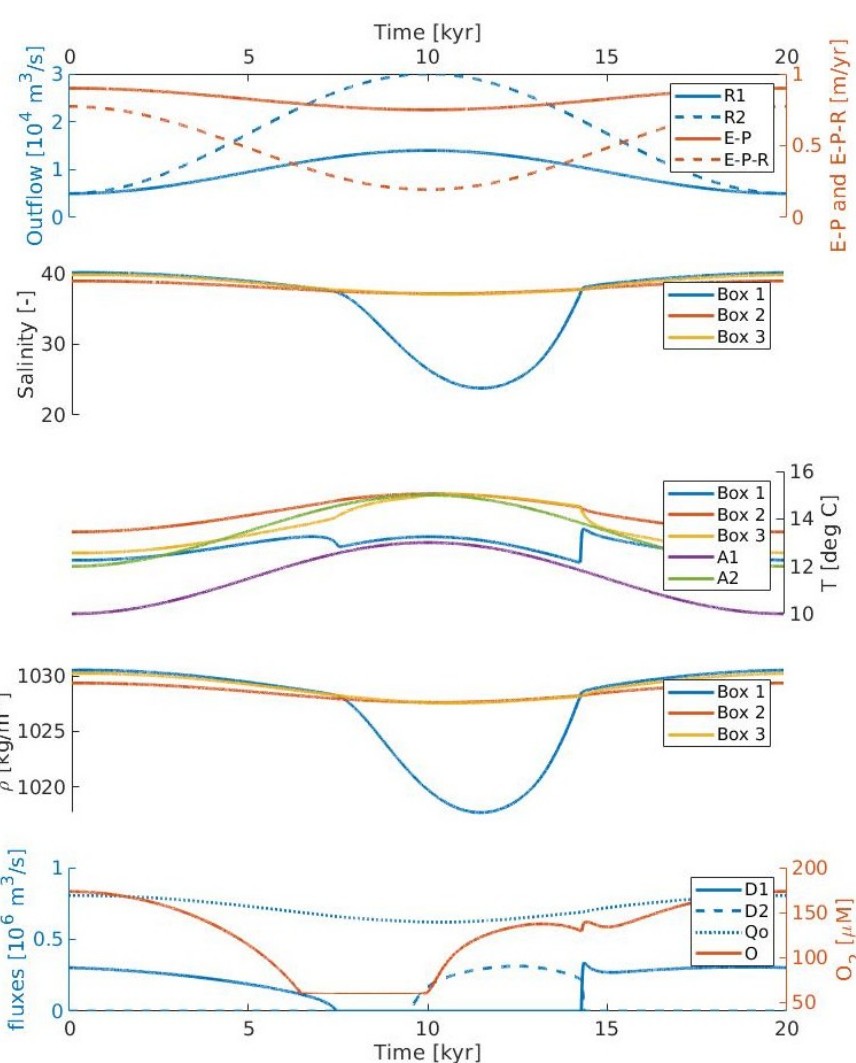

**Figure 5.** The forcing and results of the run where the freshwater budget of the margins becomes positive for a brief period. Layout of the panels is the same as in Fig. 3.



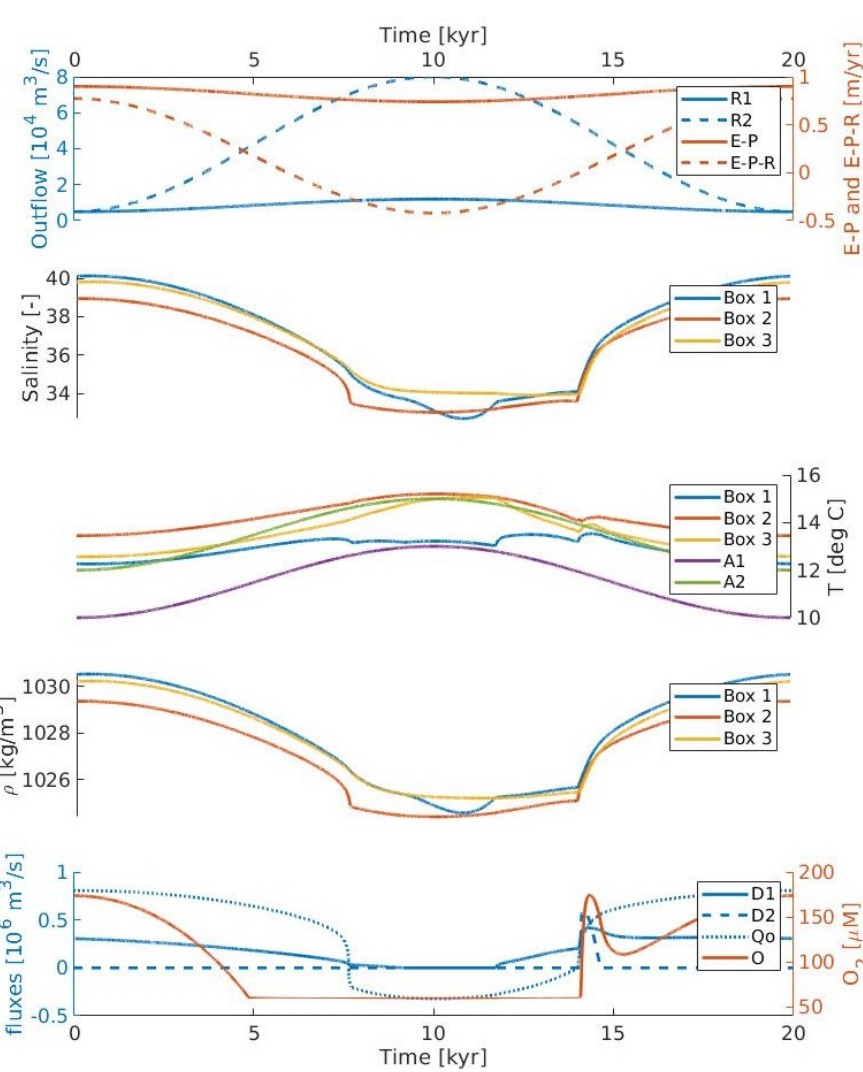

**Figure 6.** The forcing and results of the run where the freshwater budget of the whole basin becomes positive for a brief period. Layout of the panels is the same as in Fig. 3.

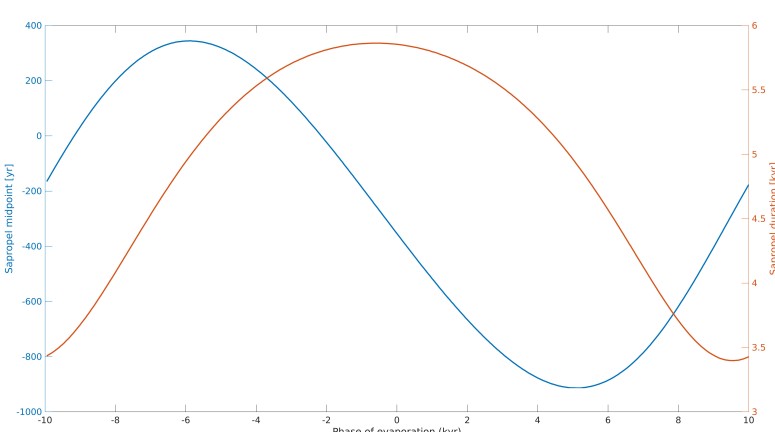

**Figure 7.** Sapropel duration (left axis) and timing of the midpoint relative to the precession minimum (on the right axis) as a function of the phase of evaporation. Apart from the phase of evaporation, the model forcing is the same as the first run in subsection 3.1





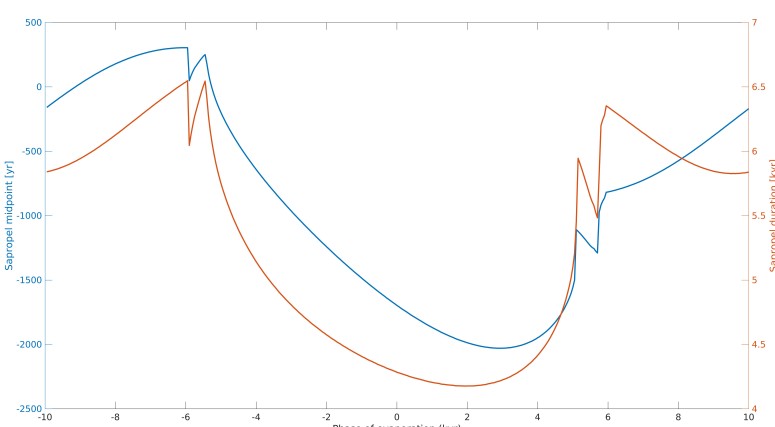

**Figure 8.** Sapropel duration (left axis) and timing of the midpoint relative to the precession minimum (on the right axis) as a function of the phase of evaporation. Apart from the phase of evaporation, the model forcing is the same as the first run in subsection 3.3