# Peer review of "The mechanism of sapropel formation in the Mediterranean Sea: Insight from long duration box-model experiments"

_Climate of the Past, 2019_

## Referee Comment (RC1) · Anonymous Referee #1 · 21 Nov 2019

The authors explore key mechanisms of sapropel formation using a simplified transient model over a full precession cycle. This is the first time such an approach has been used and I find the results very interesting, notwithstanding model limitations and caveats. The study is thorough and nicely presented, and its findings will be of interest to those working on Mediterranean palaeoclimate. I was specifically asked to review the applicability of the work for future palaeo efforts in proxy-based studies, (in short: yes, it's applicable), but I also have some comments on the modelling and minor edits for the text.

[Figure]

The study's applicability to future work is that it can help distinguish between the relative importance of different forcing mechanisms, and highlight where/how we could look for evidence of these processes. There are obviously many simplifications in the model (these are discussed), and these should be borne in mind when assessing the results. For instance, lack of seasonality and no separate boxes for the East and West basins (which each have different run-off, evaporation and ventilation regimes) means that the model could be missing important mechanisms for sapropel formation/preservation. The same goes for potential effects of meltwater pulses (from Atlantic and the EIS).I note (lines 233-235) the authors say that 'an arbitrary configuration (and number) of boxes' could be used, so why not use 6 boxes (Boxes 1-3 each for the east and west). The authors could also expound on the different DWF mechanisms (line 483), as these as significant for sapropel formation but they are not modelled here. Finally, in light of the limitations/caveats, I have an issue with the final 3 lines of the Conclusion - I don't think you can make such a strong conclusion about sapropel formation from this study.

However, on a positive note (and I do like this study), the strong agreement between the reference experiments and modern observations (eg deep water fluxes and O2 concentrations) suggests that the model is nonetheless capturing key processes for sapropel formation. The same goes for the agreement between geological proxy data and the modelled timing and duration of anoxia (and by inference, sapropel formation). I also like the investigation of switching the FW budget, both for the margins and open box, as it hints at what we could expect to see in the sediment record if such a switch occurred.

Model duration. The model is run over a full precession cycle, yet it is insolation – which includes an obliquity component – which seems to be the primary driver of long-term African monsoon variability over the Pio-Pleistocene. Modelling studies have demonstrated how obliquity forcing is significant for the African monsoon, and proxy data show the best match with local summer insolation and/or tropical insolation gradients (not all sapropels are associated with precession minima). I assume that going beyond one full

precession cycle is beyond the scope of this study, and I appreciate that just to do one full precession cycle is an advance, but some comment on this is needed, especially as the authors state (line 117) that any sine wave could be used.

Lags/phasing. The study investigates the phase & duration of sapropels relative to precession as a function of the phase of evaporation, but I think it would be more useful to investigate sapropel timing/duration as a function of the phasing of run-off. The one study they cite re: variable phase of evaporation is for the Miocene, for which we have much less understanding about individual sapropels and their associated E-P, anoxia, etc. Yet many studies have shown links between the timing +/or intensity of palaeo-monsoons and ice-sheet & North Atlantic climate variability, and run-off appears to be the primary driver of sapropel formation.

Minor technical comments

Background: A map of the Med with its seas, basins etc may be useful for newcomers to the study of this region. The text mentions the Aegean, Levantine, Adriatic, etc Line 26 & throughout: 'relatively high latitude'...I think better to refer to the more northerly parts of the basin? I wouldn't say any of the basin is at a relatively 'high latitude. Line 36: West African summer monsoon (not East). Also some clarification here that the low density surface lid is not due to direct monsoon precipitation over the basin but via run-off Line 51: clarify sapropel mid-point Line 52: perform long runs Line 125: up to 8.8 times Line 145: an efficiency Line 237: Except for the lack of a flux Line 374: reaches Line 428: delete 'thousands' – the max midpoint phase is <1000. '...up to humdreds of years' would be more accurate. Line 458: as long as a sufficiently Line 471: as accurately as possible Line 514 influencing Line 550: deep eater in the open Line 551: by reversing the freshwater budget Line 556: of a hypothetical core Line 566: add Grimm et al 2015 Figures 3-6: a-e are not labelled Figures 7-8: font size too small; reorder 'left/right axis' in the caption (wrong way around). Figure A1: can't differentiate between blue & black lines

---

## Referee Comment (RC2) · Anonymous Referee #2 · 19 Dec 2019

This manuscript uses a box model to improve our understanding of Sapropel processes in the Mediterranean Sea. By the use of a box model, the authors are able to integrate it for long enough to look at transient behavior, and changes into/out of the sapropel. The authors use a simple 3 box model, forced by atmospheric and runoff changes. The authors show that they can develop a sapropel in such a simplified model. By analyzing the model space, the authors are able to provide insights on timing, intensity, interruptions and the relative role of different forcings in combination.

This is an interesting paper, generally well written and easy to follow. Thus it adds

to our knowledge on this topic and is worth publishing. I would recommend minor revisions as there are some ways that the manuscript can be improved.

Authors – Only the institution is given. Is there no department or unit?

Introduction: Map, with circulation schematic would be useful to readers who are not familiar with the region.

Line 64: How are the later two models more advanced?

Line 71-77: The description of the box could be improved. Line 76 mentions that the Atlantic box, as well as the rivers are static boxes, yet the Atlantic box has been introduced and defined yet. Figure 1 uses subscripts in many cases, e.g. R1 while the text uses R1. Be consistent through the entire paper.

Line 78-87: Again, issues of subscripts or not. The authors mention E-P-R, but P is never defined as precipitation. And given since the box model diagram uses e (lower case), is this then a net evaporation (E-P). Basically, the whole discussion feels a little choppy without it being precisely defined. Additionally, as I was reading this paragraph, I was wondering why no equations. Now, they appear later in the manuscript, in section 2.3. But I'm not sure is the separation of the discussion and the associated equations is the best way to make things clear for the reader. The authors should at least think some more on this, and how best to clearly present their model.

Line 110-113: Found these sentences unclear. Please rewrite.

Line 126: What about outflow/runoff from the Black Sea. What is its magnitude and where is it considered in the model?

Line 167: Wouldn't a flux approach work better than simple temperature relaxation?

Line 239: Subscripts for variables

Figure 2: Listed as the second figure, but didn't find a reference to it until near the end of the paper. If so, renumber and move to where referred to.

One Sentence paragraphs: Appears many times in the paper. They are not proper English and should not be used. In all cases, it should be easy to combine them with surrounding material.

Line 289: The authors mention that the decrease in vertical density difference causes a decrease in DWF. Yet wouldn't a decrease in the vertical stratification mean that it would be easier to produce deep water formation with the same heat flux?

Line 315: Line stretches into margin

Line 321: '. . .to the DWF one of. . .' – a word seems to be missing

Line 322: What is exactly meant by 'within error'

Line 365 (and additionally later in paper): River 1 – Out of river box 1 – i.e. the river flow into the given box, not the flow of a single river

Line 374: By normal values, do you mean present day?

Experiments: As I went through the paper, I realized the authors had lots of experiments. This is good in terms of exploring the parameter space and relevant ideas. But hard to keep track of. Please add a table of experiments, listing them, giving them all an easy to follow name, and clearly listing the parameters (so that it is easy to see what is changed in each).

Line 396: decrease

Line 450: Subscripts

Line 528: The authors say that exchange through the Bosphorus is out of the scope of paper. Sure, the model can't look at the sea-level changes that lead to that connection. But in terms of impacts, the change is more runoff, which the authors can and do look at with their model. So I don't see this distinction.

Line 529: define 'within error'

Line 550: in the open. . .

Line 577: . . .system. Without it. . .

Line 578: sufficient sapropels - ??? – word(s) missing

Figure A1: I can barely see the difference between the black and blue lines. Use something more distinct. R1 should use a subscript too.

Table 1: In the descriptions, some fields uses capitals, others don't. Be consistent. Also, be careful with subscripts as elsewhere in the manuscript.

Figure 3, etc. Panels A, B, C, D, and E are not labelled.
* * *

---

## Referee Comment (RC3) · Anonymous Referee #3 · 23 Dec 2019

Review for "The mechanism of sapropel formation in the Mediterranean Sea: Insight from long duration box-model experiments"

This manuscript uses a box model to run transient simulations to investigate mechanisms of Sapropel formation in the Mediterranean Sea. The transient changes in the water cycle is based on changes in density calculated from salinity and temperature changes. Oxygen is also modelled and is used to estimate when Sapropel formation occurred. Typically modelling studies investigating Sapropel formation use a steady state or time slice approach based on defined conditions and do not consider long term

transient changes in circulation due to computational limitations. The box model presented by these authors investigate the impact of transient changes over 20,000 years so I believe this manuscript would of interest to the Mediterranean scientific community.

Although I believe that this manuscript is a novel and important study, I have some concerns. Firstly, the authors do not split the Mediterranean into Western and Eastern basins. The straits of Sicily are an important constraint on the circulation of the Mediterranean Sea and the limitation of not including this barrier to deep water circulation is barely discussed. An interesting feature of Sapropels are that they are much more dominant in the Eastern rather than Western basin so I worry about the impact of not separating them. Secondly I believe that the oxygen model is too simple. The authors use two different constant fluxes to describe oxygen consumption and these are implemented as a step function with a different consumption rate when oxygen is more than and less than 60uM. In models oxygen consumption should be proportional to oxygen concentration with either a rate constant or something like Michaelis Menton/Monod kinetics rather than the step function used here. In addition, Powley et al. (2016) show that oxygen consumption in the Mediterranean varies depending on source of the organic matter reaching the deep ocean, which ideally would be included in the oxygen model. This is important as they show that the Mediterranean has a self-regulating mechanism whereby oxygen consumption decreases when deep water formation stops due to a lower amounts of DOC reaching the deep waters. More comments concerning this can be found in the detailed comments below.

In general the written English is good and understandable but I feel that the paper is poorly organised meaning that it is hard to follow what is happening. There are methods in results and results/methods in the discussion. I suggest that the reference simulation and subsequent scenarios are introduced in the methods section with the possibility of a table detailing each simulation. In addition I suggest having separate sections for the oxygen model and building of the water cycle. Finally I suggest that the authors go through the manuscript carefully checking that all acronyms and parameter

names are clearly defined somewhere within the paper in addition to using consistent terminology for the boxes and inputs throughout. Can the authors also please cross check that all values and figures presented in the text match those in tables and figure numbers in the figure section as there were numerous time where there were inconsistencies.

Detailed comments

Section 1.2: Please can you include some of the conclusions from the modelling studies

Line 71: Please can you explicitly say which areas are in the high latitude marginal basins i.e. does this include the Adriatic and Aegean Seas.

Line 79: Similar to above please can you say the locations where D1 and D2 refer to

Line 128: What about river discharge from Europe? I assume Rhone and Elbe go into the open ocean?

Lines 122-134: You report both present day and historic values. What are you using in the your model? It is not clear to me here. You also mention that changes from Europe are not included but then talk about changes from Europe?

Section 2.3: Please include somewhere here technical details on running the model. Which method do you use to integrate forward in time, what time step was used, how frequent was the model output?

Line 145: Here you describe $c_{13}$ and $c_{23}$ as an efficiency constant but in Table 1 is described as conductivity between boxes. Please can you either add more description to the text or be consistent in descriptions

Lines 146-149: I am struggling to understand what is happening here, mostly because the processes such as D2 were not explained as mentioned above and it all seems rather abstract. What do you mean assuming the DWF in box 2 is the same as box 1?.

Please reference the sentence "D2 does not occur annually"

Lines 216-219: The consumption rates for biotic and abiotic oxygen consumption are not the same as in Table 1. I also suggest defining the acronyms for the terms in the text (i.e. biological consumption= OCB). This would also make the terms in equation 22 easier to understand as you wouldn't constantly have to refer to Table 1.

Lines 216-219: Please briefly explain the biotic and abiotic processes. Why is there no biological oxygen consumption below 60uM? Typically oxygen consumption is described using monod kinetics (i.e Vichi et al. (2015), Powley et al. (2016), Testa et al. (2014)) so that it still occurs below 60 uM but is slower. This implemented step function will likely produce the non linearality found in the model Line 217-218: Please describe how the oxygen consumption changes with river outflow.

Equation 22: Please define Rtot. It is not mentioned in the text or Table1. I assume it is total river flow which looking at the units for OCR might be in m3/s? If it it then OcR would then have to be changed to uM/yr?. It is also not clear to me why the oxygen consumption is divided by dt when over 60uM.

Line 225: Initial water temperature? Or water plus air?

Line 229: What are TA1, TA2 and T0?

Line 231: Where are the winter air temperatures taken from?

Line 233 - 265: Before my next comment I wish to say that I am not used to reading model equations in matrix format, I am used to them as ODEs. However, I found it hard to follow this section and found description of the various matrices were poorly described in some cases, for example what is matrix F or matrix M? In regards to Equation 29, if written in matrix format I would like to see in words what the equation means because as it stands I am not following what is happening and cannot check simple things like units are correct. As a more general comment, I feel it may be better to put this section describing the matrix equations at the beginning of section 2.3 and

then explain what how the fluxes and parameters are calculated afterwards.

Section 3: I suggest explaining the different runs in the methods section and potentially having a table describing each simulation and the model setup used.

Lines 268-276, 332-339 etc: The forcings applied to the runs should be described in the methods section, not here

Line 320-321: Which value is observational and which is from the model? Please reference the observational data

Line 343-345: "we find a sapropel from t=2900 years to 6500 years". I don't see this in figure 4E as O2 looks low from around 7000 to 12000 years. In fact to me figure 4E looks remarkable similar to the reference run and I would suggest that you may look zoom into the mark around 60uM for oxygen concentration. This also means that the conclusion that the addition of atmospheric temperature variability in the model has a large impact on Sapropel formation could be wrong.

Line 365: I can't see evidence of a positive freshwater budget in Figure 5A.

Line 365-366: "the maximum outflow of river 1 is increased from 6.7 .103 to 1.4 .104 ." In the reference simulation the maximum outflow of European rivers (I assume R1?) was 1.2 x 104, so I don't understand: a) where 6.7 comes from and b)how this is different from the reference simulation. I can't see any noticeable differences in R1 between Fig 3A and Fig 5A either

Line 416: What do you mean by irregulaties?

Lines 448-454: This should be in the methods (or maybe results), not opening the discussion.

Lines 456-464: The model timestep is not mentioned in methods so it rather comes out of the blue discussing it here. Also be specific in the writing. Temporal resolution of what? Model outputs or model timestep?

[Figure]

Line 470: "Main hypothesis". What is your main hypothesis? This is not stated clearly either here or in the introduction.

Line 483: Please describe the two different mechanisms

Line 509: "A simple threshold analysis will not suffice either". Please explain what you mean be a threshold analysis. Surely the method you are implementing with oxygen is a threshold analysis?

Line 525-527: In the introduction you say Sapropels are caused by African monsoon whereas here you are saying that other mechanisms can cause them. Please clarify in the introduction and go more into depth of different mechanisms and hypotheses for Sapropel formation.

Line 529: Please can you quantify "within values", i.e. explicitly compare values in the literature with what you found.

Line 537: What do you mean by strait efficiency?

Line 539: Please explain what you mean by alternative regimes

Figures:

Figure 2: Please move to end of paper in line with when it is mentioned in the text.

Figures 3-6: Please label panels with A,B C, D and E. Please explain for what boxes E-P and E-P-R represent. It would be nice rather than use box 1, 2 etc, you could use marginal , open ocean etc and then it would match up with the text. I also suggest using the same scaling for axes across figures to make comparison between figures easier, for example the scale on the axis for outflow changes in Figure 6A compared to Figure 3A.

Minor Comments

Line 122: suggest putting R2 in brackets after box 2 for clarification

Line 218: Add additional bracket after 2016

Line 295+304: Fig 3E rather than Fig. 3D?

Line 301: I assume "it " is oxygen concentration? Be specific

Line 273: Suggest putting R2 after Nile outflow for clarification.

Line 354: increase rather than increases

Line 370: Fig 5E rather than D?

Line 380: "Deep water oxygen largely behaves as the total DWF". I do not understand this sentence. Please rephrase.

Line 422: "subsection 3.2" The caption for Fig 7 says subsection 3.1

Line 456: annual resolution of what? model outputs?

References

Powley, H.R., Krom, M.D., Van Cappellen, P. (2016) Circulation and oxygen cycling in the Mediterranean Sea: Sensitivity to future climate change. Journal of Geophysical Research:Oceans, 121, 8230-8247, doi: 10.1002/2016JC012224.

Testa, J. M., Y. Li, Y. J. Lee, M. Li, D. C. Brady, D. M. Di Toro, W. M. Kemp, and J. J. Fitz-patrick (2014), Quantifying the effects of nutrient loading on dissolved O2 cycling and hypoxia in Chesapeake Bay using a coupled hydrodynamic–biogeochemical model, J Mar Syst, 139, 139158, doi: 10.1016/j.jmarsys.2014.05.018

Vichi, M., T. Lovato, P. Lazzari, G. Cossarini, M. E. Gutierrez, G. Mattia, S. Masina, W. J. McKiver, N. Pinardi, C. Solidoro, L. Tedesco, and M. Zavatarelli (2015), The Bio-geochemical Flux model (BFM): Equation Description and USer Manual. BFM version 5.1. BFM Report Series N. 1, Release 1.1, August 2015 Bologna, Italy, http://bfm-community.eu, pp 104

---

## Author Comment (AC1) · 31 Jan 2020

We thank the reviewers for their appreciation for our general approach and for three thoughtful reviews. Below we respond to the issues that are being raised.

Response to reviewer 1

Comment: The study's applicability to future work is that it can help distinguish between the relative importance of different forcing mechanisms, and highlight where/how we could look for evidence of these processes. There are obviously many simplifications in

the model (these are discussed), and these should be borne in mind when assessing the results. For instance, lack of seasonality and no separate boxes for the East and West basins (which each have different run-off, evaporation and ventilation regimes) means that the model could be missing important mechanisms for sapropel formation/preservation.

Response: We agree, we will add this to the discussion. Seasonality is mostly important as a driving mechanism for the DWF. Including seasonality would require separate intermediate water boxes (increasing complexity), while for the oxygenation deep water only the amount of DWF and mixing with the overlying water mass is truly relevant. Furthermore, we would have to make assumptions regarding the annual variability of the forcing parameters (river outflow and evaporation), which are not well constrained for geological history. We therefore decided to parameterize the seasonal variability, by calculating a yearly averaged DWF flux based on winter temperatures. This allows us to study the fundamental mechanisms of sapropel formation. Perhaps one needs to turn to OGCMs to study the role of seasonal variation. We will add a discussion on the lack of East/West boxes. Also see our comment below regarding lines 233-235.

Comment: The same goes for potential effects of meltwater pulses (from Atlantic and the EIS).

Response: We agree that melt water pulses likely affect sapropel formation, but do not consider them to be of first order importance. During many sapropels, melt water pulses did not occur. In future applications of this model where a specific sapropel/interval is studied, drivers such as melt water pulses should of course be included, but this is beyond the scope of this paper. We will add this explanation to section 4.3 (around line 473) in the revised manuscript.

Comment: I note (lines 233-235) the authors say that 'an arbitrary configuration (and number) of boxes' could be used, so why not use 6 boxes (Boxes 1-3 each for the east and west).

Response: The aim of a conceptual model is to capture the first order aspects of a process with a minimal setup. As noted by the reviewer, the current setup does this. Doubling the number of boxes would also double the number of forcing parameters and equations, all of which add uncertainty to the model (quantitative reconstruction do not exist for most of these parameters). Moreover, the complexity quickly increases, making it much harder to test and describe the parameter space, and identify key mechanisms. The main purpose of this specific sentence was to point out that the matrix-vector formulation adopted in our paper is applicable equally well to a larger number of boxes. See also our response to a similar question by reviewer 3.

Comment: The authors could also expound on the different DWF mechanisms (line 483), as these as significant for sapropel formation but they are not modelled here.

Response: These two mechanisms are mixing of the water at margins during winter storms, which then cascades to the deep basin, and open ocean convection. This is explained in the Introduction (lines 26-31) and we will refer this in the discussion in the revised manuscript. See also our response to a similar question by reviewer 3.

Comment: Finally, in light of the limitations/caveats, I have an issue with the final 3 lines of the Conclusion - I don't think you can make such a strong conclusion about sapropel formation from this study.

Response: While noting that the start of this sentence makes it clear that this concerns, first and foremost, a suggestion about sapropel formation based on model results, we point out that the conclusion is in line with previous studies (Grant et al., 2016, and references therein) that found that sea surface temperature fluctuations occurred during at least some of the recent sapropels. Here we quantify this effect. With the new formula where oxygen consumption is linearly dependent on oxygen concentration (as suggested by reviewer 3), the role of temperature is more prominent. In the revised manuscript we will add that this conclusion is supported by previous studies.

Comment: However, on a positive note (and I do like this study), the strong agreement between the reference experiments and modern observations (eg deep water fluxes and O2 concentrations) suggests that the model is nonetheless capturing key processes for sapropel formation. The same goes for the agreement between geological proxy data and the modelled timing and duration of anoxia (and by inference, sapropel formation). I also like the investigation of switching the FW budget, both for the margins and open box, as it hints at what we could expect to see in the sediment record if such a switch occurred.

Response: Thank you, we agree.

Comment: Model duration. The model is run over a full precession cycle, yet it is insolation – which includes an obliquity component – which seems to be the primary driver of long-term African monsoon variability over the Pio-Pleistocene. Modelling studies have demon- strated how obliquity forcing is significant for the African monsoon, and proxy data show the best match with local summer insolation and/or tropical insolation gradients (not all sapropels are associated with precession minima). I assume that going beyond one full precession cycle is beyond the scope of this study, and I appreciate that just to do one full precession cycle is an advance, but some comment on this is needed, especially as the authors state (line 117) that any sine wave could be used.

Response: We are mainly interested in the response of the system to a transient forcing, it is not our aim to reconstruct the exact conditions during specific time intervals. For an individual sapropel, adding an obliquity component would effectively slightly modulate the frequency and amplitude of the forcing. Since the model is not very sensitive to the exact frequency of the forcing, and we already extensively tested the parameter space in terms of amplitude, a simple (20 kyr) sine wave suffices as forcing. We will add a comment to this extent. Also note that since obliquity does not have a harmonic relation with precession, the modulation would not have the same effect on every sapropel. It likely affects the thickness of a sapropel for example, but the effect may work both ways when comparing different sapropels. Again, this will be added to a new version. Note that line 117 mentions "any temporal variation could be used...".

[Figure]

This implies that the same model code could be applied to different settings/scenarios, or that actual reconstructions (based on sediment cores) or output of other models could be used to force the model rather than just a sine wave.

Comment: Lags/phasing. The study investigates the phase & duration of sapropels relative to precession as a function of the phase of evaporation, but I think it would be more useful to investigate sapropel timing/duration as a function of the phasing of run-off. The one study they cite re: variable phase of evaporation is for the Miocene, for which we have much less understanding about individual sapropels and their associated E-P, anoxia, etc. Yet many studies have shown links between the timing +/or intensity of palaeo-monsoons and ice-sheet & North Atlantic climate variability, and run-off appears to be the primary driver of sapropel formation.

Response: Run-off and evaporation are the only transient forcings in the presented model runs, therefore shifting run-off for example 2 kyrs forward in time gives the exact same wave shape as shifting evaporation 2 kyrs backwards in time. The only difference would be that the waveform would be shifted by 4 kyrs. Since we are primarily interested in the transient response rather than the absolute timing, we consider the presented experiments to be sufficient. We will add this description (and note that that the timing of increased river outflow may also vary) to paragraph 3.4.

Minor technical comments

Comment: Background: A map of the Med with its seas, basins etc may be useful for newcomers to the study of this region. The text mentions the Aegean, Levantine, Adriatic, etc

Response: Thank you for this suggestion, we will add a map to the revised manuscript.

Comment: Line 26 & throughout: 'relatively high latitude'. . .I think better to refer to the more northerly parts of the basin? I wouldn't say any of the basin is at a relatively 'high latitude.

Response: We do consider the difference in latitude compared to the rest of the basin to be important, as the difference in temperature is a major driver of the circulation in the Mediterranean Sea. We change the sentence to: "During winter, in the northerly parts of the basin, situated at relatively high latitude, cold and dry winds induce a further density increase, which may lead to the formation of deep water (Schroeder et al., 2012)."

Comment: Line 36: West African summer monsoon (not East). Also some clarification here that the low density surface lid is not due to direct monsoon precipitation over the basin but via run-off

Response: We will change this to "African summer monsoon" (in accordance with Grant et al., 2016, Rohling et al., 2015, etc.). Hennekam et al. (2015) finds that during S1 Nile discharge was likely not predominantly controlled by the West African summer monsoon. We added a clarification that the low density lid is not due to direct monsoon precipitation over the basin.

Comment: Line 51: clarify sapropel mid-point

Response: The average of the top and bottom age, we will add this explanation to in the revised manuscript.

Comment: Line 428: delete 'thousands' – the max midpoint phase is <1000. '. . .up to humdreds of years' would be more accurate.

Response: With a higher evaporation variability amplitude, the midpoint phase can shift by more than 1000 years, we will add this remark to the revised manuscript.

Comments: Line 52: perform long runs Line 125: up to 8.8 times Line 145: an efficiency Line 237: Except for the lack of a flux Line 374: reaches Line 458: as long as a sufficiently Line 471: as accurately as possible Line 514 influencing Line 550: deep eater in the open Line 551: by reversing the freshwater budget Line 556: of a hypothetical core Line 566: add Grimm et al 2015 Figures 3-6: a-e are not labelled Figures 7-8:

font size too small; reorder 'left/right axis' in the caption (wrong way around). Figure A1: can't differentiate between blue & black lines

Response: We acknowledge all these points and will correct them in the revised manuscript.

Response to reviewer 2

Comment: Authors – Only the institution is given. Is there no department or unit? Response We will add "Department of Earth Sciences" in the revised manuscript.

Comment: Introduction: Map, with circulation schematic would be useful to readers who are not familiar with the region.

Response: Thank you for this suggestion, we will add a map to the revised manuscript. The lack of this was pointed out by Reviewer 1 also. We do not consider a circulation schematic useful. Figure 1 in the manuscript gives an overview of how the circulation is abstracted in the model, while the articles we refer to (for example Pinardi et al., 2015) give a clear and complete description of the present circulation of the Mediterranean Sea.

Comment: Line 64: How are the later two models more advanced?

Response: In the revised manuscript we will change this sentence to "...and more recently, using a regional ocean model forced by output from a dedicated global climate model experiment, Mikolajewicz (2011) and Adloff et al. (2011)."

Comment: Line 71-77: The description of the box could be improved. Line 76 mentions that the Atlantic box, as well as the rivers are static boxes, yet the Atlantic box has been introduced and defined yet. Figure 1 uses subscripts in many cases, e.g. R1 while the text uses R1. Be consistent through the entire paper.

Response: We will correct this in the revised manuscript.

Comment: Line 78-87: Again, issues of subscripts or not.

Response: We will correct this in the revised manuscript.

Comment: The authors mention E-P-R, but P is never defined as precipitation. And given since the box model diagram uses e (lower case), is this then a net evaporation (E-P). Basically, the whole discussion feels a little choppy without it being precisely defined. Additionally, as I was reading this paragraph, I was wondering why no equations. Now, they appear later in the manuscript, in section 2.3. But I'm not sure is the separation of the discussion and the associated equations is the best way to make things clear for the reader. The authors should at least think some more on this, and how best to clearly present their model.

Response: We will define P as precipitation and clearly define e/net evaporation. The separation between the description of the model and the presentation of the equations was done on purpose, after careful consideration and based on previous experience with this type of papers. We expect that most paleoceanographers and biogeologists are not primarily interested in the exact equations, but ought to be able to find in the paper a detailed description of the model. See also our response to a similar question by reviewer 3.

Comment: Line 110-113: Found these sentences unclear. Please rewrite.

Response: We will rewrite the sentences in the revised manuscript, along the lines of: "We therefore abstract the circulation to an open surface/intermediate box, a marginal surface/intermediate box and a deep water box, all with constant volumes. While the formation of deep water itself is a seasonal process, we parametrize the seasonal variability by calculating an annually averaged DWF flux. We know that DWF occurs every year during present winters. However, deep water would not form with annual average conditions, we therefore assume perpetual winter conditions."

Comment: Line 126: What about outflow/runoff from the Black Sea. What is its magnitude and where is it considered in the model?

Response: For sapropels during which there is exchange through the Bosphorus strait, the exchange is not constant through time, and also depends on the inflow of Mediterranean water into the Black Sea. Opening or closing of the strait prior to, or during, a sapropel may impact the circulation. When the sill becomes deep enough to allow for a two layer exchange, a large amount of saline water would flow into the Black Sea (following the same principle as at the Gibraltar Strait), thereby causing extra relatively fresh water to flow out into the Mediterranean Sea. During some sapropels, the strait may have been closed. Furthermore, there is very little data regarding the exchange opening and closing of the Bosphorus Strait prior to the most recent opening (approximately 11 ka), perhaps with the exception of the Pontian (which is beyond the scope of this paper). Consequently, we do not include exchange with the Black Sea. For the cases where there was a steady outflow of fresh water (or an exchange that can be parametrized as such) this could indeed be seen as an extra fresh water source for the margins. We have already tested this effect by varying the river outflow into the margins. We will add this to section 4.3 in the revised manuscript. Please also see our response to a related comment by Reviewer 3.

Comment: Line 167: Wouldn't a flux approach work better than simple temperature relaxation?

Response: In a previous version of the model we used a constant flux (of 5 W/m2). With a normal circulation this gives similar results, but when the fresh water budget of the margins approaches zero, and the circulation (almost) stops, the results are not realistic. In this situation the margins become almost completely isolated from the rest of the basin, causing a massive temperature drop that doesn't stop until the circulation starts again. In reality this temperature drop would be limited by the atmospheric temperature, hence a relaxation is more realistic. We will add this explanation to section 2.3 in the revised manuscript.

Comment: Line 239: Subscripts for variables

Response: We will correct this in the revised manuscipt

Comment: Figure 2: Listed as the second figure, but didn't find a reference to it until near the end of the paper. If so, renumber and move to where referred to

Response: We will renumber and move the figure in the revised manuscript.

Comment: One Sentence paragraphs: Appears many times in the paper. They are not proper English and should not be used. In all cases, it should be easy to combine them with surrounding material.

Response: We will combine one sentence paragraphs with the surrounding material in the revised manuscript.

Comment: Line 289: The authors mention that the decrease in vertical density difference causes a decrease in DWF. Yet wouldn't a decrease in the vertical stratification mean that it would be easier to produce deep water formation with the same heat flux?

Response: Indeed, when the deep water has a density that is only slightly above that of the surface water, a given temperature decrease of the surface water would sooner lead to an instable situation and to deep water formation. However, our text at this point refers to a situation in which the overlying water mass (already) has a higher density. Then we expect that a decrease in the density difference will cause a decrease in DWF.

Comment: Line 315: Line stretches into margin Line 321: '. . .to the DWF one of. . .' – a word seems to be missing

Response: This should be " ...to the DWF of one of...". We will correct this in the revised manuscript.

Comment: Line 322: What is exactly meant by 'within error'

Response: The timing of the sapropel in the model corresponds to the timing found in the cited article within the error margin of the dating of the sediment core. We will add this to the revised manuscript.

Comment: Line 365 (and additionally later in paper): River 1 – Out of river box 1 – i.e. the river flow into the given box, not the flow of a single river Response We will correct this in the revised manuscript.

Comment: Line 374: By normal values, do you mean present day?

Response: Yes, we will change this to "present day values" in the revised manuscript.

Comment: Experiments: As I went through the paper, I realized the authors had lots of experiments. This is good in terms of exploring the parameter space and relevant ideas. But hard to keep track of. Please add a table of experiments, listing them, giving them all an easy to follow name, and clearly listing the parameters (so that it is easy to see what is changed in each).

Response: Attached to this response are two tables that give a clear overview of all the presented runs. These we would include in a revised version of the paper.

Comment: Line 396: decrease Line 450: Subscripts Response We will correct this in the revised manuscript.

Comment: Line 528: The authors say that exchange through the Bosphorus is out of the scope of paper. Sure, the model can't look at the sea-level changes that lead to that connection. But in terms of impacts, the change is more runoff, which the authors can and do look at with their model. So I don't see this distinction.

Response: To this point we have already responded in the above. See comment to Line 126.

Comment: Line 529: define 'within error

Response: We will correct this in the revised manuscript.

Comment: Line 550: in the open. . . Line 577: . . .system. Without it. . .

Response: We will correct these errors in the revised manuscript.

Comment: Line 578: sufficient sapropels - ??? – word(s) missing

Response: This should be "sufficiently long sapropels". We will correct this in the revised manuscript.

Comment: Figure A1: I can barely see the difference between the black and blue lines. Use something more distinct. R1 should use a subscript too.

Response: We will correct this in the revised manuscript.

Comment: Table 1: In the descriptions, some fields uses capitals, others don't. Be consistent. Also, be careful with subscripts as elsewhere in the manuscript.

Response: We will correct this in the revised manuscript.

Comment: Figure 3, etc. Panels A, B, C, D, and E are not labelled.

Response: We will correct this in the revised manuscript. See figures 1 and 2 in the attachment for an example.

Response to reviewer 3

Comment: Although I believe that this manuscript is a novel and important study, I have some concerns. Firstly, the authors do not split the Mediterranean into Western and Eastern basins. The straits of Sicily are an important constraint on the circulation of the Mediterranean Sea and the limitation of not including this barrier to deep water circulation is barely discussed. An interesting feature of Sapropels are that they are much more dominant in the Eastern rather than Western basin so I worry about the impact of not separating them.

Response: The aim of a conceptual model is to capture the first order aspects of a process with a minimal setup. The current setup does this. Doubling the number of boxes would also double the number of forcing parameters and equations, all of which add uncertainty to the model (quantitative reconstruction do not exist for most of these parameters). Moreover, the complexity quickly increases, making it much

harder to test and describe the parameter space, and identify key mechanisms. This is definitely something we might look into in a future study, but consider it important to first understand the behavior of the a semi-enclosed basin with a gateway before studying what is essentially a second order system. We expect that the effect in the Eastern basin of doing so is similar to the difference between a first order and second order filter: a larger shift of the midpoint (larger group delay), and likely a higher sensitivity. Any resonances in the system are also expected to become more prominent (since the resonant frequencies are now amplified twice). We will discuss this in the revised manuscript.

Comment: Secondly I believe that the oxygen model is too simple. The authors use two different constant fluxes to describe oxygen consumption and these are implemented as a step function with a different consumption rate when oxygen is more than and less than 60uM. In models oxygen consumption should be proportional to oxygen concentration with either a rate constant or something like Michaelis Menton/Monod kinetics rather than the step function used here. In addition, Powley et al. (2016) show that oxygen consumption in the Mediterranean varies depending on source of the organic matter reaching the deep ocean, which ideally would be included in the oxygen model. This is important as they show that the Mediterranean has a self-regulating mechanism whereby oxygen consumption decreases when deepwater formation stops due to a lower amounts of DOC reaching the deep waters. More comments concerning this can be found in the detailed comments below.

Response: We have now also tried a formulation in which the amount of oxygen consumption is linearly dependent on oxygen concentration, instead of a step function. Although the shape of the dependency between consumption and concentration is different from that implied by Monod kinetics, this new set-up does capture the basic notion of a proportionality to oxygen concentration. The results are very similar when considering everything below 60uM a sapropel, see figures 1 and 2 in the attachment. Fig. 1 is a run with exactly the same forcing as the reference run in the manuscript,

[Figure]

and Fig. 2 a run with the exact same forcing as the temperature variability run in the manuscript. Note that in the temperature variability run the minimum in bottom water oxygen is lower and the interval with oxygen below 60uM therefore somewhat longer. We agree that the feedback related to DOC described by the reviewer is an interesting mechanism to study, however we find that this is beyond the scope of this study. Note that with the new formula oxygen consumption already responds to changes in DWF: as the circulation slows down, less oxygen is supplied to the deep water, causing the oxygen concentration to drop, thereby reducing oxygen consumption. Furthermore, as a follow-up study we are adapting this water model to a different setting, where we combine it with a nutrient model that includes DOC. While the nutrient model provides further insights, the resulting oxygen concentrations are very similar to the ones found when using both the step function and the linear dependence we use here. In conclusion, we can adjust the oxygen consumption formula to a linear dependence, this does not significantly affect the main findings. See the attached figures for runs of the first two regimes using this new oxygen consumption formula.

Comment: In general the written English is good and understandable but I feel that the paper is poorly organised meaning that it is hard to follow what is happening. There are methods in results and results/methods in the discussion. I suggest that the reference simulation and subsequent scenarios are introduced in the methods section with the possibility of a table detailing each simulation. In addition I suggest having separate sections for the oxygen model and building of the water cycle. Finally I suggest that the authors go through the manuscript carefully checking that all acronyms and parameter names are clearly defined somewhere within the paper in addition to using consistent terminology for the boxes and inputs throughout. Can the authors also please crosscheck that all values and figures presented in the text match those in tables and figure numbers in the figure section as there were numerous time where there were inconsistencies.

Response: We will improve the consistency in the revised manuscript as suggested by

the reviewer (also see responses do the detailed comments below). Two tables with the forcing parameters of each presented simulation are attached to this response. These we would include in a revised version.

Comment: Section 1.2: Please can you include some of the conclusions from the modelling studies Response We will add this to the revised manuscript.

Comment: Line 71: Please can you explicitly say which areas are in the high latitude marginal basins i.e. does this include the Adriatic and Aegean Seas. Response Yes, this does include the Adriatic and Aegean Seas. We will add this to the revised manuscript.

Comment: Line 79: Similar to above please can you say the locations where D1 and D2 refer to

Response: D1 occurs in box 1, and D2 in box 2, so the same locations as these boxes. We will add this explanation to the revised manuscript.

Comment: Line 128: What about river discharge from Europe? I assume Rhone and Elbe go into the open ocean?

Response: We assume that the reviewer meant the Rhone and Ebro. In the current model setup, the Gulf of Lyon is considered to be a marginal deep water formation area, thereby included in box 1. The Ebro would indeed flow into the open ocean, although it is relatively unimportant since its present day discharge is only one fifth of that of the Nile.

Comment: Lines 122-134: You report both present day and historic values. What are you using in the your model? It is not clear to me here. You also mention that changes from Europe are not included but then talk about changes from Europe?

Response: We intended to state that Amies et al. (2019) do not include change from Europe. We will replace "this model" by "their study" to explain this more clearly. We do indeed include changes from Europe.

[Figure]

Comment: Section 2.3: Please include somewhere here technical details on running the model. Which method do you use to integrate forward in time, what time step was used, how frequent was the model output?

Response: The equations are integrated numerically simply by the forward Euler method taking appropriately small time steps. We use a time step of 1 year, except when testing the effect of the time step in section 4.2. The curves shown in the figures are built up of the output at every time step. We missed multiplication by dt in equations 27-29 (only in the manuscript, in the matlab code the equations are correct), e.g., (27) should read T(t+1) = T(t)+(G+N+H) Âů T(t) ÂůWÂů dt. This will be corrected in the revised manuscript.

Comment: Line 145: Here you describe c13 and c23 as an efficiency constant but in Table 1 is described as conductivity between boxes. Please can you either add more description to the text or be consistent in descriptions.

Response: We will change "conductivity" to "efficiency" in the revised manuscript.

Comment: Lines 146-149: I am struggling to understand what is happening here, mostly because the processes such as D2 were not explained as mentioned above and it all seems rather abstract. What do you mean assuming the DWF in box 2 is the same as box 1?

Response: The deep water formation mechanisms are explained in the introduction, we will refer to the introduction in lines 146-149 in the revised manuscript. We assume that for both of these mechanisms the amount of DWF in a year is linearly dependent on the density difference between the boxes in question.

Comment: Please reference the sentence "D2 does not occur annually"

Response: This can be phrased more accurately as " deep convection in the Levantine basin (represented by D2) does not occur every year (Gertman et al, 1994; Pinardi et al., 2015).

Comment: Lines 216-219: The consumption rates for biotic and abiotic oxygen consumption are not the same as in Table 1. I also suggest defining the acronyms for the terms in the text (i.e. biological consumption= OCB). This would also make the terms in equation 22 easier to understand as you wouldn't constantly have to refer to Table 1.

Response: With the new oxygen consumption formula, as suggested by reviewer 3, these parameters are ommited. We agree that defining acronyms in the text would improve readability, and will do so where necessary in the revised manuscript.

Comment: Lines 216-219: Please briefly explain the biotic and abiotic processes. Why is there no biological oxygen consumption below 60uM? Typically oxygen consumption is described using monod kinetics (i.e Vichi et al. (2015), Powley et al. (2016), Testa et al.(2014)) so that it still occurs below 60 uM but is slower. This implemented step function will likely produce the non linearality found in the model.

Response: As explained in the above we have now made oxygen consumption dependent on oxygen concentration, thereby replacing the step function. See also our response to the more general comment. This change yields largely the same results when considering everything below 60uM a sapropel. See the attached figures for preliminary results using this new formula. The step function caused the oxygen concentration to stop decreasing at 60uM, other non-linear behavior is not related to the oxygen consumption formula.

Comment: Line 217-218: Please describe how the oxygen consumption changes with river outflow.

Response: Oxygen consumption increases linearly with river outflow as can be gleaned from Equation (22) in combination with Table 1. In this equation the Ocx represent the three types of oxygen consumption. OcR is the coefficient that, upon multiplication with the river discharge, gives the contribution to the rate of oxygen consumption thought related to rivers. This more explicit explanation will be added to the revised manuscript.

Comment: Equation 22: Please define Rtot. It is not mentioned in the text or Table1. I assume it is total river flow which looking at the units for OCR might be in m3/s? If it it then OcR would then have to be changed to uM/yr?. It is also not clear to me why the oxygen consumption is divided by dt when over 60uM.

Response: Rtot is the sum of R1 and R2, we will correct this in the revised manuscript. With the new oxygen consumption formula the units of the parameters are different, we will carefully check all the units in the revised manuscript.

Comment: Line 225: Initial water temperature? Or water plus air?

Response: The initial temperature of the dynamic boxes. The Atlantic ocean, atmosphere and river boxes are static. The dynamic boxes are boxes 1, 2 and 3, as defined in the methods.

Comment: Line 229: What are TA1, TA2 and T0?

Response: These are the temperatures of boxes A1, A2, and 0. The abbreviations of the boxes is shown in Fig. 1. We will add this explanation to the revised manuscript.

Comment: Line 231: Where are the winter air temperatures taken from?

Response: These are intended as present day values at the precession maximum. winter SST ranges from∼10 °C in the northwest to 15–16 °C in the southeast (Naval Oceanography Command, 1987). We will add this to the revised manuscript.

Comment: Line 233 - 265: Before my next comment I wish to say that I am not used to reading model equations in matrix format, I am used to them as ODEs. However, I found it hard to follow this section and found description of the various matrices were poorly described in some cases, for example what is matrix F or matrix M? In regards to Equation 29, if written in matrix format I would like to see in words what the equation means because as it stands I am not following what is happening and cannot check simple things like units are correct. As a more general comment, I feel it may be better to put this section describing the matrix equations at the beginning of section 2.3 and

then explain what how the fluxes and parameters are calculated afterwards.

Response: We acknowledge that the matrix equations are likely difficult to understand for many readers, this is why we have decided to explain all the equations in words before showing them and explaining them mathematically. We will more clearly state that the fluxes (such as F2,1) are elements of the matrix F. We think it is more logical to explain the equations of all the fluxes before introducing the matrix calculations, since the fluxes are used in the matrix calculations.

Comment: Section 3: I suggest explaining the different runs in the methods section and potentially having a table describing each simulation and the model setup used.

Response: We consider the runs to be results, since we describe part of the parameter space. We therefore prefer to keep the explanation of the runs in the results. We will add a table describing each run to the results (see tables 1 and 2 in the attachment). We will add a paragraph to the methods where we explain how we tested the parameter space (e.g. identify different regimes in the model).

Comment: Lines 268-276, 332-339 etc: The forcings applied to the runs should be described in the methods section, not here

Response: In the results we describe part of the parameter space. Which part we describe, i.e., what forcing is applied, is in part determined from/inspired by the results of the preceding experiments. It felt as unnatural to make the strict separation suggested by the Reviewer

Comment: Line 320-321: Which value is observational and which is from the model? Please reference the observational data

Response: This sentence will be changed to "The deep water flux at the precession maximum (3 Âů 105 m3/s) is somewhat lower than what is found in observational data (1.6 Âů 106 m3/s, Pinardi et al., 2015), although comparable to the DWF one of the Eastern sub basins (Pinardi et al., 2015)".

Comment: Line 343-345: "we find a sapropel from t=2900 years to 6500 years". I don't see thisin figure 4E as O2 looks low from around 7000 to 12000 years. In fact to me figure 4Elooks remarkable similar to the reference run and I would suggest that you may lookzoom into the mark around 60uM for oxygen concentration. This also means that the conclusion that the addition of atmospheric temperature variability in the model has a large impact on Sapropel formation could be wrong.

Response: The sentence "we find a sapropel from t=2900 years to 6500 years" was erroneously not changed to 7900-11500 after we shifted the timing of the start of the run by 5000 years. We will correct this in the revised manuscript. With the step function formula for oxygen consumption there, the effect of temperature variability is present, but indeed not very prominent in the figures. With the new oxygen formula (where oxygen consumption is linearly dependent on oxygen concentration) the effect of atmospheric temperature variability is more prominent and also more clearly visible.

Comment: Line 365: I can't see evidence of a positive freshwater budget in Figure 5A.

Response: We currently only show the freshwater budget for the entire basin (which does not change sign in this run), to keep the graphs more readable. We will add the freshwater budget of box 1 separately to the figures in the revised manuscript, figures 1 and 2 in the attachment illustrate this (note that these runs do not have a positive fresh water budget).

Comment: Line 365-366: "the maximum outflow of river 1 is increased from 6.7 .103 to 1.4 .104." In the reference simulation the maximum outflow of European rivers (I assume R1?)was 1.2 x 104, so I don't understand: a) where 6.7 comes from and b)how this is different from the reference simulation. I can't see any noticeable differences in R1between Fig 3A and Fig 5A either

Response: This was an error, and will be corrected in the revised manuscript. The 6.7 is the minimum outflow of R1. The difference between 1.2*104 and 1.4*104 is relatively small, and therefore hard to see in the graph.

Comment: Line 416: What do you mean by irregulaties?

Response: The occurrence of multiple local minima. We will add this to the revised manuscript.

Comment: Lines 448-454: This should be in the methods (or maybe results), not opening the discussion.

Response: Point taken. We will move this to the methods in the revised manuscript.

Comment: Lines 456-464: The model timestep is not mentioned in methods so it rather comes out of the blue discussing it here. Also be specific in the writing. Temporal resolution of what? Model outputs or model timestep?

Response: We will mention the model time step in the methods in the revised manuscript, as well as that the model output is generated at every time step. See also in the above (comment to section 2.3).

Comment: Line 470: "Main hypothesis". What is your main hypothesis? This is not stated clearly either here or in the introduction.

Response: With this we refer to the commonly accepted scenario for sapropel formation as it is explained This is explained in the introduction (lines 34-37), we will explain that we use this as our main hypothesis in the revised manuscript and refer to the introduction when we mention it on line 470.

Comment: Line 483: Please describe the two different mechanisms

Response: This is described on lines 23-31. These two mechanisms are mixing of the water at margins during winter storms, which then sinks to the deep water, and open ocean convection. We will refer to the introduction in the discussion in the discussion in the revised manuscript.

Comment: Line 509: "A simple threshold analysis will not suffice either". Please explain what you mean be a threshold analysis. Surely the method you are implementing with

oxygen is a threshold analysis?

Response: We will rephrase this part to something along the lines of: "A simple threshold analyses is not ideal either, as the cut-off level can have major impact on both timing and duration, while a clear definition is not readily available. Furthermore, even when the threshold is defined, this method would not be usable for sapropelic marls, which are thought to be the result of the same process, but do not share the same chemical composition. We partly avoid this problem by not considering the midpoint of the sapropel (when assuming a certain oxygen threshold, see subsection 4.5), but also the full wave form (e.g. which intervals could be sapropelic with a slightly different forcing). In the sedimentary record this is generally not possible, since the non-sapropelic intervals do not record all parameters and are often bioturbated. So while our approach can't be related applied to the sedimentary record, it does give insight into factors influence sapropel timing."

Comment: Line 525-527: In the introduction you say Sapropels are caused by African monsoon whereas here you are saying that other mechanisms can cause them. Please clarify in the introduction and go more into depth of different mechanisms and hypotheses for Sapropel formation.

Response: We here mention that sapropel S1 may have been triggered by sea level rise, this does not exclude monsoon intensity variability as the main cause. We will add this to the introduction.

Comment: Line 529: Please can you quantify "within values", i.e. explicitly compare values in the literature with what you found.

Response: In the model with the new oxygen consumption formula in the run with variable air temperature (Fig. 2 in the attachment) the interval where deep water oxygen is below 60uM lasts from 7.5 to 11.6 kyr (with maximum insolation at 10 kyr), Grant et al. (2016) found that S3 lasted from 80.8-85.8 ka, with an uncertainty of $2.0 \pm 0.9$ kyr, and the maximum in summer inter-tropical insolation gradient at 82.5 ka. We will change

"...within error..." to "...within dating uncertainty..." in the revised manuscript.

Comment: Line 537: What do you mean by strait efficiency?

Response: The magnitude of the density driven flow at the strait for a given density difference, i.e, the coefficient of proportionality between volume transport and density difference. We will add this to the methods.

Comment: Line 539: Please explain what you mean by alternative regimes

Response: Parts of the parameter space where one or more fluxes change direction (as a result of a change in freshwater budget of either part of the basin, or the entire basin), as presented in Figures 5 and 6. We will add a similar description to the revised manuscript.

Comment: Figure 2: Please move to end of paper in line with when it is mentioned in the text.

Response: We will correct this in the revised manuscript.

Comment: Figures 3-6: Please label panels with A,B C, D and E. Please explain for what boxes E-P and E-P-R represent. It would be nice rather than use box 1, 2 etc, you could use marginal , open ocean etc and then it would match up with the text.

Response: We will label panels A, B, C, and D in the revised manuscript, see figures 1 and 2 in the attachment. We prefer to use box 1, 2, etc., to prevent covering a larger part of the graph with the legend. The numbering of the boxes is clearly defined in the methods and Figure 1.

Comment: I also suggest using the same scaling for axes across figures to make comparison between figures easier, for example the scale on the axis for outflow changes in Figure 6A compared to Figure 3A.

Response: Currently the graphs are already very small to accommodate for the axis titles and space between the graphs. We prefer to use as much of the available space

as possible. For example, note setting all the axis limits to the same value would imply setting the minimum salinity to approximately 20, which would make the graphs very hard to read.

Comment: Line 122: suggest putting R2 in brackets after box 2 for clarification

Response: We will add R2 in brackets after box 2 in the revised manuscript.

Comment: Line 218: Add additional bracket after 2016

Response: This will be corrected in the revised manuscript

Comment: Line 295+304: Fig 3E rather than Fig. 3D? Line 301: I assume "it " is oxygen concentration? Be specific

Response: Yes, figure 3E and "it" is oxygen concentration. We will correct this, and revise this part to reflect the results of the new oxygen consumption formula in the revised manuscript.

Comment: Line 273: Suggest putting R2 after Nile outflow for clarification.

Response: We will add R2 after Nile outflow in the revised manuscript.

Comment: Line 354: increase rather than increases

Response: This will be corrected in the revised manuscript

Comment: Line 370: Fig 5E rather than D?

Response: Yes, this will be corrected in the revised manuscript.

Comment: Line 380: "Deep water oxygen largely behaves as the total DWF". I do not understand this sentence. Please rephrase.

Response: We will rephrase. Deep water oxygen largely correlates with DWF.

Comment: Line 422: "subsection 3.2" The caption for Fig 7 says subsection 3.1

Response: This will be corrected in the revised manuscript.

Comment: Line 456: annual resolution of what? model outputs?

Response: The time step. We will change the sentence to "The time step of one year..." in the revised manuscript.

Additional references used in our response Naval Oceanography Command, 1987. U.S. Navy climatic study of the Mediterranean Sea. Naval Oceanography Command Detachment, Asheville, North Carolina (342 pp.).

Please also note the supplement to this comment:
https://www.clim-past-discuss.net/cp-2019-128/cp-2019-128-AC1-supplement.pdf

**Supplement:**

**The mechanism of sapropel formation in the Mediterranean Sea: Insight from long duration box-model experiments**

Jan Pieter Dirksen[1] and Paul Th. Meijer[1]

[1]Utrecht University

**Correspondence:** Jan Pieter Dirksen (j.p.dirksen@uu.nl)

**Table 1.** Forcing parameters that are the same for all runs

| Parameter | Value | units |
|-----------|-------|-------|
| R1min | $5 \cdot 10^3$ | $m^3/s$ |
| R2min | $3 \cdot 10^3$ | $m^3/s$ |
| emax | 0.9 | $m/yr$ |
| TA1min | 10 | $\deg C$ |
| TA2min | 12 | $\deg C$ |

**Table 2.** Forcing parameters that vary between runs.

| Run name | R1max $(m^3/s)$ | R2max $(m^3/s)$ | emin $(m/yr)$ | TA1max $(\deg C)$ | TA2max $(\deg C)$ |
|----------|-----------------|-----------------|---------------|-------------------|-------------------|
| Reference run | $1.2 \cdot 10^4$ | $3 \cdot 10^4$ | 0.75 | 10 | 12 |
| Temperature variability run | $1.2 \cdot 10^4$ | $3 \cdot 10^4$ | 0.75 | 13 | 15 |
| fwb1 run | $1.2 \cdot 10^4$ | $3 \cdot 10^4$ | 0.75 | 13 | 15 |
| fwbtot run | $1.4 \cdot 10^4$ | $8 \cdot 10^4$ | 0.74 | 13 | 15 |

[Figure]

**Figure 1.** The forcing and results of the reference run. (A) The model forcing, with the river outflow on the left axis and the evaporation on the right axis. (B)-(D) For each box respectively the salinity, temperatures, and densities. (E) The relevant fluxes (left axis) and the deep water oxygen concentration (right axis)

[Figure]

**Figure 2.** The forcing and results of the temperature variability run. Layout of the panels is the same as in Fig. 1. Note that the minimum in oxygen is slightly lower than in the reference run in Fig. 1E